# Distribution patterns of fungal community diversity in the dominant tree species *Dacrydium pectinatum* and *Vatica mangachapoi* in tropical rainforests

Kepeng Ji,[1,2] Yaqing Wei,[1,3] Xin Wang,[1,4] Yu Liu,[5] Rui Sun,[1,3] Yuwu Li,[6] Guoyu Lan[1,3]

**ABSTRACT** Plant microbial communities are shaped by plant compartments, but the patterns of fungal communities in aboveground and belowground compartments, and which environmental factors can affect them, remain unknown. Here, to address this research gap, high-throughput sequencing technology was performed to investigate the diversity of fungal communities in leaves' and roots' compartments of *Dacrydium pectinatum* and *Vatica mangachapoi* from Hainan Island of China. Fungal communities in leaves and roots exhibited significant differences. *Eurotiomycetes* (16.57%) and *Dothideomycetes* (45.57%) were predominantly found in leaves, while *Agaricomycetes* (36.53%) dominated in roots. Compared to the roots, the leaf compartments had higher α-diversity. According to the Mantel test, soil pH mainly influenced roots, while the main driving factors for leaves were rainfall and temperature. The proportion of dispersal-limited processes in rhizoplane (76.67%) and root endosphere (73.81%) were greater than that in leaf epiphytic (62.38%) and leaf endophytic (68.1%), driven by ectomycorrhizal fungi with known dispersal limitations. In summary, the compositions of the leaf and root fungal communities of both endangered tree species differed, partly driven by environmental factors unique to each compartment. Our results provide valuable theoretical and practical insights for preserving tropical tree species.

**IMPORTANCE** Understanding the assembly of microbial communities across different compartments is a prerequisite for harnessing them to enhance plant growth. Our findings reveal significant differences in fungal community structures between the root and leaf compartments. Compared to the roots, the leaf compartments exhibited higher α-diversity. While soil pH mainly influenced fungal communities in the roots, the primary drivers for the leaves were rainfall and temperature. The dispersal-limited processes of fungal communities in the roots were greater than those in the leaves, primarily influenced by mycorrhizal fungi. These findings demonstrate compartment-specific plant-microbe interactions and environmental responses, offering actionable insights for conserving tropical tree species through habitat optimization (e.g., soil pH management) and dispersal corridor preservation. This compartment-aware perspective enhances our ability to leverage microbial functions to improve the resilience of endangered trees in the face of climate change.

**KEYWORDS** fungal community, *Dacrydium pectinatum*, *Vatica mangachapoi*, community assembly, diversity, environmental factors

Microorganisms are ubiquitous in nature, and the microbial communities present in and on plants are an extremely important part of this diversity. Microorganisms can inhabit various parts of plants, including internal tissues, surfaces, and other compartments, collectively described as the plant microbiota (1). The two

**Peer Reviewer** Daolong Xu, Jiangnan University, Wuxi, China

Address correspondence to Yuwu Li, liyuwu@qau.edu.cn, or Guoyu Lan, langyrri@163.com.

Kepeng Ji and Yaqing Wei contributed equally to this article. The author order was determined both alphabetically.

The authors declare no conflict of interest.

main compartments for plant microorganisms are the phyllosphere (the aboveground compartments primarily refer to the leaf) and the rhizosphere (the belowground parts of plants) (2, 3). Phyllosphere microbiota can be epiphytic or endophytic (4). Plant roots have three continuous and fine-scale compartments: the rhizosphere, the rhizoplane, and the endosphere. The rhizosphere is composed of approximately 1 mm of soil that adheres tightly to roots and is not easily removed, while the rhizoplane consists of the root surface and finer soil that clings directly to the root surface (5). Plants and microorganisms together form a "plant holobiont," and there has always been interaction between them (6, 7). Numerous research studies claimed microorganisms linked to plants are conducive to their health (3, 8). Specifically, plants and microorganisms exhibit some adaptations and interactions that enable plants to enhance growth, survive in harsh environments, and overcome adverse conditions (9, 10). However, research on microbiota associated with rare and endangered plants in tropical rainforests is relatively scarce.

Analyzing how different ecological processes influence microbial community structure is crucial for revealing the mechanisms of community assembly and for predicting shifts in community composition (11). They involve two key processes: one is deterministic factors rooted in niche theory, primarily encompassing heterogeneity selection and homogeneity selection, and the other is stochastic processes derived from neutral theory, which includes dispersal limitation, homogenizing dispersal, and drift (12). Both abiotic and biotic factors significantly affect the plant microbiome. These factors include plant identity, host selection, geographic location, and soil properties (13–17). In the aboveground compartment, geographical variation shapes the composition of phyllosphere fungi by temperature and rainfall (14). Certainly, the influence of the host cannot be ignored either (18). In the belowground compartment, while different geographical locations significantly impact the rhizosphere fungal communities, the primary influencing factor is soil pH (19). Besides these deterministic factors, fungal communities are also affected by dispersal limitations (20). Clearly, the response of different fungal groups in aboveground and belowground compartments to changing climate or environmental variables may be inconsistent (21). Therefore, studying how fungal communities vary across different habitats on a regional scale contributes to predicting the effects of environmental variation and future climate change (22) and to exploring resilient protection strategies for plant aboveground and belowground systems (23).

The tropical rainforest of Hainan is an important part of China's and even the world's tropical rainforests, and it is one of the hotspots for global biodiversity conservation (24). The diverse ecosystems within Hainan's tropical rainforest nurture an incredibly rich diversity of plants, providing habitats for many rare, endangered, and endemic species of plants. *Vatica mangachapoi* serves as an important indicator species for tropical rainforests and belongs to the Dipterocarpaceae family. However, in recent years, due to indiscriminate cultivation and deforesting, its population has been severely threatened, leading to its classification as a second-level nationally protected plant. Meanwhile, *Dacrydium pectinatum*, a rare and endangered tree species in the Podocarpaceae family, is the only representative species of this genus in China. As a dominant and constructive species in the mountain rainforests of Hainan, it is only found in mountain rainforests at 700–1300 m altitude on Hainan Island, China (25). However, the microbial community compositions, diversity, and assembly patterns of important tropical protected tree species, such as *V. mangachapoi* and *D. pectinatum*, and the significance of various assembly processes remain unknown. We investigated the fungal communities in the aboveground and belowground compartments of *V. mangachapoi* and *D. pectinatum*, which is of great significance for understanding how rare and endangered plants tackle the challenges of climate change (26).

We aim to analyze the diversity, community composition, and assembly patterns of fungi between both the aboveground and belowground compartments of *V. mangachapoi* and *D. pectinatum*. Additionally, we elucidate the spatial characteristics and influence

factors of fungal community structures of different regions by incorporating various environmental factors. Our study provides important theoretical and practical insights for preserving tropical tree species. The following hypotheses are tested. (i) Fungal community compositions and diversities are influenced by plant compartments, plant identity, and geographic location. (ii) The aboveground fungal communities are primarily driven by temperature and rainfall, as leaf fungal communities exhibit high responsiveness to climate change (21). In contrast, soil pH is the primary driver in belowground communities. (iii) Dispersal limitation is the key factor influencing fungal communities of the aboveground and belowground compartments (20, 27).

## MATERIALS AND METHODS

### Description of study area and site selection

For *V. mangachapoi*, we selected four locations in Bawang, Diaoluo, Jianfeng, and Wanning on Hainan Island. Meanwhile, three sites in Diaoluo, Jianfeng, and Wuzhi on Hainan Island were chosen for *D. pectinatum* (Fig. S1). These sites are all distributed in the core area of the tropical rainforest. Three plots located approximately 5–15 km apart were chosen for each site. From each plot, we sampled three trees that were spaced roughly 100 m apart. Thus, we sampled a total of 63 trees. Specifically, there are 27 *D. pectinatum* and 36 *V. mangachapoi*. We collected leaf, root, and topsoil samples from the same tree (Fig. S2). For each compartment, the samples from the three trees were combined to produce a composite sample. Therefore, there were 105 samples in total from 5 compartments. For each sampling location, we gathered mean monthly rainfall and temperature data from the National Meteorological Information Center (https://www.data.cma.cn).

### Sample collection and analysis

Leaf samples were gathered from each tree at a height of 12 m and a distance of 2 m from the trunk. In addition, root and topsoil samples were collected from a location 0.5 m away from the trunk, at depths ranging from 5 to 20 cm below the soil surface. Approximately 300 g of mature and healthy leaf and 250 g of fine root (≤2 mm in diameter) were taken from the same tree (28). A portion of leaf samples and all topsoil were used for physicochemical property analysis. Detailed information about the physicochemical properties of leaf and topsoil was presented (Tables S1 and S2), and the detailed analysis methods are detailed in Methods S1.

To obtain leaf surface microbiota samples, leaves were placed in a sterile buffered solution (0.1 M potassium phosphate, 0.1% glycerol, 0.15% Tween 80, and pH 7.0). We conducted ultrasonication for 1 min at a frequency of 40 kHz and shook the mixture at 200 rpm for 4 min. We repeated this procedure three times. Finally, we filtered the solution using a vacuum filtration device and stored the filter paper in a −80°C freezer (29, 30). The filtered leaf samples were washed with 70% alcohol and then used for sequencing of endophytic microorganisms.

We collected the roots and followed three steps to collect rhizosphere soil, rhizoplane soil, and root-associated microorganisms. First, the soil attached to the roots was obtained by vigorous shaking to collect the rhizosphere soil. Second, the remaining roots were placed in a buffer solution, sonicated for 1 min in an ultrasonic instrument, and centrifuged for 1 min at 3,000 rpm. This process was repeated three times, after which the supernatant was discarded, and the precipitated soil was collected as rhizoplane soil. Third, the treated roots were repeatedly washed three times with 70% ethanol and sterile water and then dried using sterile cotton to extract root-associated microorganisms (31). These three samples were stored at −80°C for sequencing.

## DNA extraction and PCR amplification

We extracted fungal community genomic DNA from samples using the FastDNA spin kit for soil. All types of samples were operated according to the same kit. The specific operation process was shown in Supporting Information Methods S2. We used PCR primers ITS1F (5′-CTTGGTCATTTAGAGGAAGTAA-3′) and ITS2R (5′-GCTGCGTTCTTCATCGAT GC-3′) (32). The conditions for PCR amplification included an initial denaturation step at 95℃ for 3 min. This was followed by 35 cycles consisting of denaturation at 95℃ for 30 s, annealing at 55℃ for 30 s, and extension at 72℃ for 45 s. The process concluded with a final extension at 72℃ for 10 min. Then, the resultant PCR products were combined, and subsequent recovery was achieved through electrophoresis on a 2% agarose gel. Finally, we purified the samples using the PCR clean-up kit. Equimolar paired-end sequencing (2 × 250 bp) was performed for the purified amplicons by the Illumina PE300 platform in Shanghai Majorbio.

## Bioinformatics analysis

The raw FASTQ files were demultiplexed, quality filtered, and merged using fastp (version 0.19.6) (33, 34). The operation was as follows: the terminal bases of reads with a quality value less than 20 were filtered out, a window of 50 base pairs was set, and reads containing any "N" bases were deleted. Double-end reads were merged into one sequence according to the overlap (maximum overlap length does not exceed 10 base pairs; mismatch rate in the overlap region of merged sequences does not exceed 0.2). Sequences that did not meet this standard were removed. Samples were identified using the barcode and primer located at the sequence ends, and the sequence direction was adjusted accordingly. The barcode had to match perfectly, while for the primer, up to two mismatches were allowed.

Operational taxonomic units (OTUs) were clustered with a similarity of no less than 97% using UCHIME (version 7.2) (35). The sequence count for samples was rarefied to the minimum sample size with a uniform count of 30,101. The taxonomy of sequences was identified by the RDP classifier (version 2.2), with a matching threshold set at 70% in the UNITE fungal identification ITS database (36, 37). Please see Table S5 for the specific OTU reports at different taxonomic levels.

## Statistical analysis

The observed richness (Sobs) is an index that reflects community α-diversity, representing the actual observed OTU of richness (4). We analyzed the α-diversity index (OTU richness) by the *vegan* package (38) and the "wilcox.test" of the Wilcoxon rank-sum test was conducted in the *ggpubr* package to compare results between two and more groups in R (version 4.2.3). The community structure was analyzed by principal coordinate analysis based on the Bray-Curtis distance matrix and analyzed with PERMANOVA of the *vegan* package with 999 permutations (39).

To investigate the potential driving factors of the Sobs diversity index, the relative abundance of *Eurotiomycetes* in leaves and *Agaricomycetes* in roots, a nonlinear regression model was used for evaluation by using the "geom_smooth" function of the R package ggplot2. The impact of soil properties, climate factors, and leaf properties on the fungal communities of different compartments was studied by using the "mantel_tes" function of the ggcor package (40). Fast expectation-maximization microbial source tracking (FEAST) to reveal the source of fungal communities was performed by the FEAST package (41). FUNGuild predicted the fungal functions of the five compartments (42).

We utilized the beta nearest taxon index (βNTI) from the iCAMP package in R to determine phylogenetic β-diversity by quantifying ecological processes of community assembly (12). Values of |βNTI| > 2 indicate that the community assembly is primarily driven by deterministic processes, which can be further classified into homogeneous selection (βNTI < −2) and heterogeneous selection (βNTI > 2). While values of |βNTI|

< 2 indicate that community assembly is mainly dominated by neutral processes (43). Besides, the Raup-Crick index based on Bray-Curtis dissimilarity ($RC_{bray}$) was used to further delineate stochastic processes. $|RC_{bray}| > 0.95$ represents either homogenizing dispersal ($RC_{bray} < 0.95$) or limiting dispersal ($RC_{bray} > 0.95$) driving community change. $|\beta NTI| < 2$ and $|RC_{bray}| < 0.95$ represent drift (44).

## RESULTS

### Diversity patterns of fungal community

Overall, the γ-diversity of fungi in the aboveground (leaf epiphytic and endophytic) was much higher than that in the belowground compartments. Among the five compartments, leaf epiphyte fungi exhibited the highest γ-diversity, while the root endosphere had the lowest (Fig. 1A). The fungal α-diversity of leaf epiphytes was also the highest. Clearly, the observed OTU richness in the aboveground compartments was significantly higher than that in the belowground compartments (Fig. 1B). The α-diversity was not influenced by plant identity (Fig. 1C), but the fungal α-diversity of leaf epiphytes of *V. mangachapoi* ($P < 0.05$) and *D. pectinatum* ($P < 0.001$) varied significantly across different geographic locations, with the higher α-diversity observed in leaf samples collected from Jianfeng and Bawang, while α-diversity in the belowground compartments was little affected by geographic location (Fig. 1D and E). Particularly, the fungal α-diversity of leaf epiphytes was higher than endophytes. The analysis of variance demonstrated the geographic location and plant compartment shaped the fungal community OTU richness significantly, while plant identity did not (Table S3). In conclusion, geographic location was the main driving factor affecting the fungal α-diversity in the aboveground compartment.

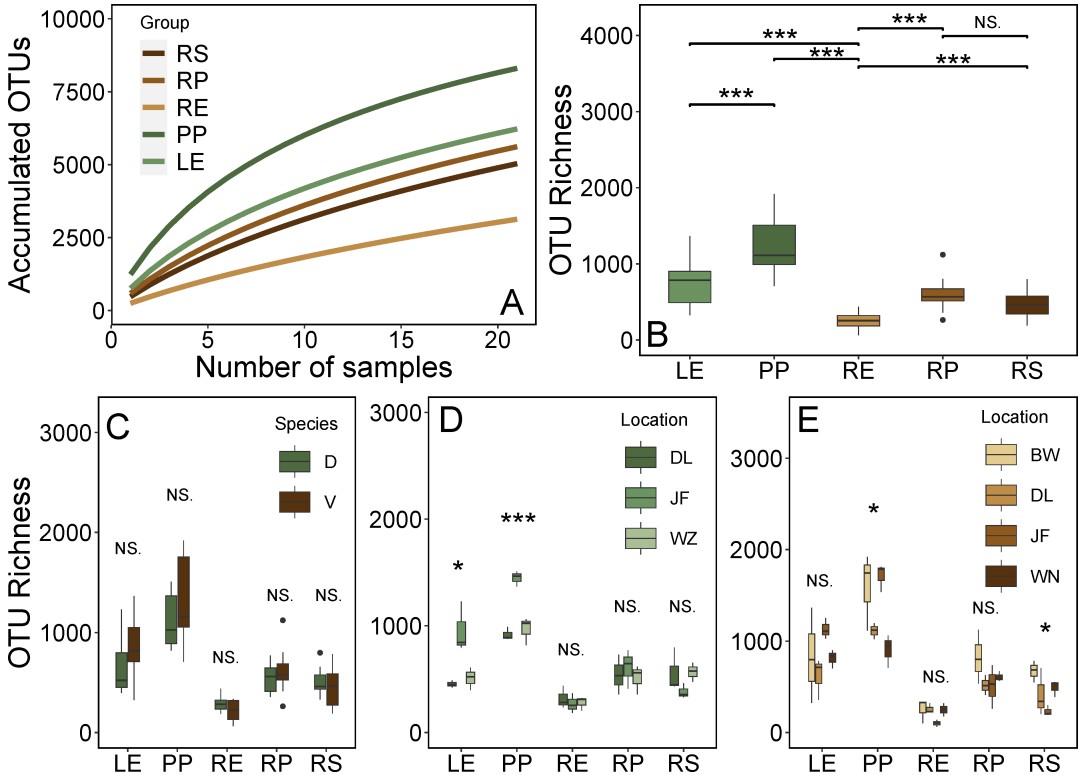

**FIG 1** Diversity of fungal community in *D. pectinatum* and *V. mangachapoi*. (A) Accumulated OTUs (γ-diversity) for all samples in five compartments. (B) Mean OTU richness (α-diversity) of fungal communities for five compartments. (C) Comparison of OTU richness of five compartments of *D. pectinatum* and *V. mangachapoi*. (D) Comparison of OTU richness of *D. pectinatum* across five compartments in three locations. (E) Comparison of OTU richness of *V. mangachapoi* across four compartments in four locations. Abbreviation: LE, leaf endophytic; PP, leaf epiphytic; RP, rhizoplane; RS, rhizosphere; RE, root endosphere; D, *D. pectinatum*; V, *V. mangachapoi*; DL, Diaoluo; JF, Jianfeng; WZ, Wuzhi; BW, Bawang; and WN, Wanning; Significance level: *$P < 0.05$; **$P < 0.01$; and ***$P < 0.001$.

## The taxon compositions

Overall, the dominant fungi exhibited differences between plant compartments, both above and below ground, and the same was true between the two tree species (Fig. 2A and B; Fig. S3). Ascomycota predominated in the aboveground compartment, while Ascomycota and Basidiomycota dominated the belowground compartment (Fig. 2A). Additionally, *Eurotiomycetes* and *Dothideomycetes* were predominantly present in aboveground compartments, with *Eurotiomycetes* being more abundant in *D. pectinatum* (Fig. 2B; Fig. S3A). Conversely, *Agaricomycetes* were mainly present in belowground compartments, with a relatively higher abundance in *V. mangachapoi* (Fig. 2B; Fig. S3B). However, only a tiny part of *Agaricomycetes* were observed in aboveground compartments (Fig. 2B). FUNGuild analysis demonstrated a substantial presence of ectomycorrhizal fungi in the belowground compartments, while such fungi were absent in the aboveground compartments (Fig. 2C). The proportions of pathotrophic–saprotrophic and pathotrophic fungi were enriched in the aboveground compartments compared to the belowground compartments, whereas symbiotrophic and symbiotrophic–saprotrophic fungi exhibited the opposite pattern (Fig. 2C). Approximately 37% of *Dothideomycetes* were classified as pathogens, and 23% of *Agaricomycetes* were classified as mycorrhizal fungi (Fig. 2D). The influence of plant compartment ($R^2 = 0.126$, $P < 0.001$), plant identity ($R^2 = 0.029$, $P < 0.001$), and geographic location ($R^2 = 0.102$, $P < 0.001$) on the fungal community composition of all samples was significant (Fig. 3A). The principal coordinates analysis (PCoA) ordination indicated that the fungal communities clustered according to aboveground and belowground compartments, with fungal communities of leaf clustering by plant identity (Fig. 3A). Compared to belowground compartments, geographic location and plant identity mainly shaped fungal composition in the aboveground compartments (Fig. 3B; Table S4). At a lower taxonomic level,

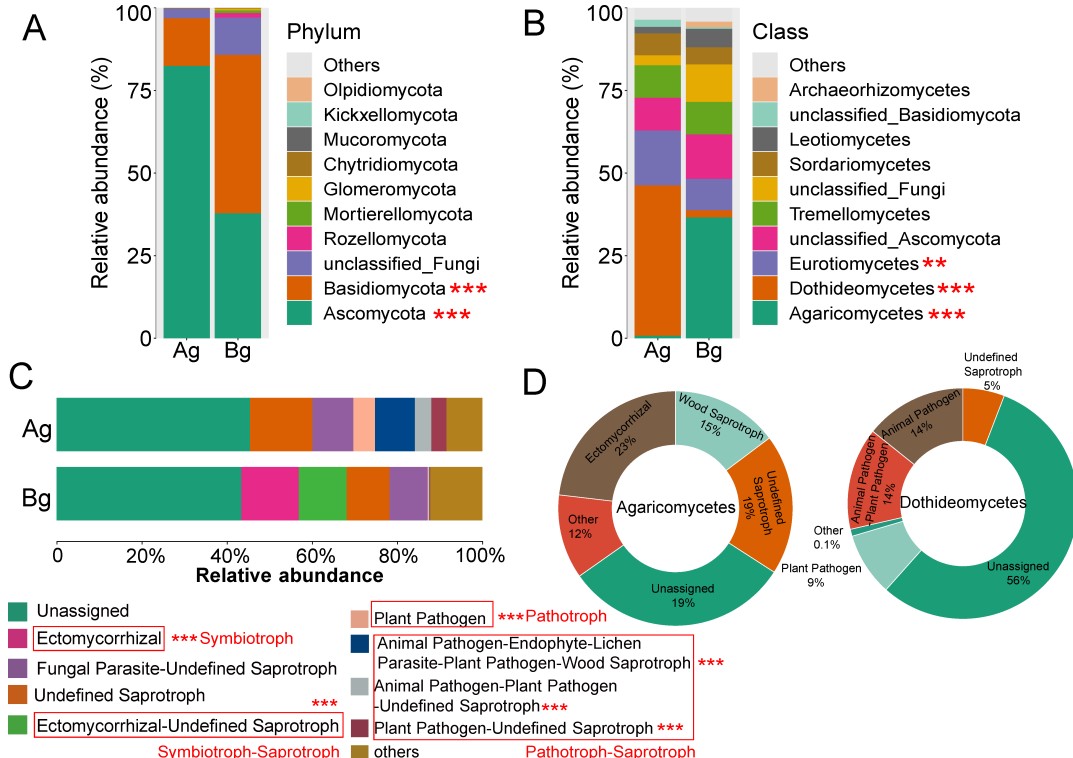

**FIG 2** The fungal community compositions and the composition of the fungal community functional groups inferred by FUNGuild for *D. pectinatum* and *V. mangachapoi*. (A) Top 10 of the phylum level. (B) Top 10 of the class level. (C) Relative abundance of fungal functions. (D) Relative abundance of fungal functions of the dominant fungal class. Abbreviation: Ag: aboveground compartment and Bg: belowground compartment. Significance level: *$P < 0.05$; **$P < 0.01$; and ***$P < 0.001$.

551 fungal genera and 814 fungal species were shared between the aboveground and underground compartments (Fig. S4A and B). At the genus level, *Saitozyma* (5.56%) and *Cladosporium* (3.96%) were the dominant taxa, while *Saitozyma* sp. (5.67%) was the dominant taxa at the species level (Fig. S4C and D). FEAST results indicated that although the sources of leaf-related fungi in aboveground compartments were mostly unknown, over 40% of epiphytic and endophytic fungal communities originated from each other. However, there was minimal contribution from belowground compartments to aboveground fungal communities (Fig. S5). In conclusion, geographic locations and plant identities significantly impact fungal community composition in the aboveground compartments.

## The associations of environmental factors

The Mantel test results showed rainfall and temperature mainly shape the leaf-associated aboveground fungal communities, with α-diversity significantly correlated with rainfall and temperature (Fig. 4A and B). Soil pH crucially shaped the compositions of endosphere and rhizosphere fungal communities (Fig. 4C and E). Linear regression models also confirmed that pH, rainfall, and temperature were key driving factors affecting the diversity of aboveground leaf-associated fungal communities (Fig. 5A). As rainfall increased, the diversity of leaf-associated fungal communities increased ($R^2 = 0.26$, $P < 0.001$), as did temperature ($R^2 = 0.34$, $P < 0.001$) and pH ($R^2 = 0.19$, $P < 0.05$) (Fig. 5A). Therefore, the diversity of leaf-associated fungal communities was higher in Jianfeng and Bawang than other locations (Fig. 5A). *Eurotiomycetes* in leaf-associated fungal communities were significantly negatively correlated with pH ($R^2 = 0.17$, $P < 0.01$), Mg ($R^2 = 0.13$, $P < 0.05$), temperature ($R^2 = 0.17$, $P < 0.01$), and rainfall ($R^2 = 0.10$, $P < 0.05$). Therefore, for *D. pectinatum,* a higher relative abundance of *Eurotiomycetes* in leaf-associated fungal communities was observed (Fig. 5B; Table S2). *Agaricomycetes* in root-associated fungal communities were significantly positively correlated with pH ($R^2 = 0.23$, $P < 0.001$), but significantly negatively correlated with AN ($R^2 = 0.16$, $P < 0.001$), WC ($R^2 = 0.30$, $P < 0.001$), and SOM ($R^2 = 0.15$, $P < 0.01$). Therefore, the relative abundance of *Agaricomycetes*

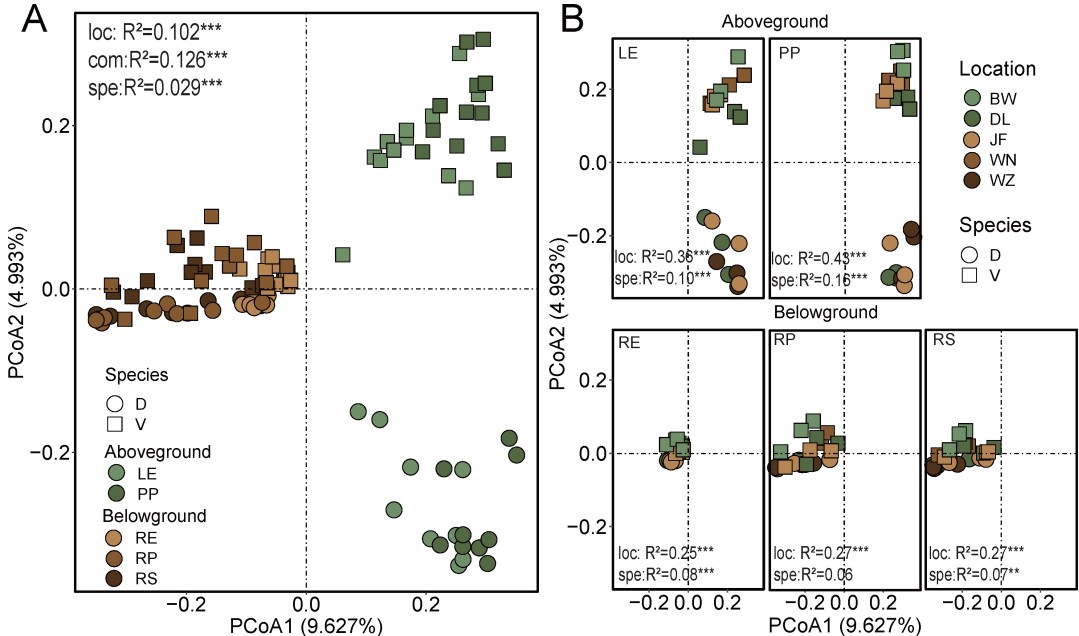

**FIG 3** PCoA of taxonomic similarity based on Bray-Curtis distances for fungal community compositions at the OTU level. (A) Five compartments. (B) *D. pectinatum* and *V. mangachapoi* in five geographical locations. Abbreviation: LE, leaf endophytic; PP, leaf epiphytic; RP, rhizoplane; RS, rhizosphere; RE, root endosphere; D, *D. pectinatum*; V, *V. mangachapoi*; DL, Diaoluo; JF, Jianfeng; WZ, Wuzhi; BW, Bawang; and WN, Wanning; Significance level: *$P < 0.05$; **$P < 0.01$; and ***$P < 0.001$.

in root-associated fungal communities was higher in *V. mangachapoi* compared to *D. pectinatum* (Fig. 5C; Table S2).

## The community assembly and function

The result indicated that most βNTI values were between −2 and 2, which revealed the stochastic processes dominantly mediating the fungal community assembly in the five plant compartments (Fig. 6A). Furthermore, the βNTI values of root-associated fungal communities were significantly higher than those of leaf-associated communities, indicating a stronger stochasticity in the belowground compartments. The result indicated the mean values of Raup-Crick in five compartments were higher than 0.95, which revealed that dispersal limitation dominates the fungal community assembly in the five plant compartments (Fig. 6B). This is seen for the dominant classes Dothideomycetes and Agaricomycetes in the aboveground and belowground compartments (Fig. 6C; Fig. S6). The dispersal limitation process in root-associated fungal communities was consistently higher than in leaf-associated communities (except for the rhizosphere), while the proportion of dispersal limitation for *Dothideomycetes* was higher than *Agaricomycetes* (Fig. 6C; Fig. S6).

## DISCUSSION

### Aboveground fungal community diversity exceeds that of belowground

Our results indicated fungal diversity was primarily influenced by compartment and geographic location, but not by plant identity. However, previous studies have shown that host traits influence fungal diversity to different degrees (45, 46). Climatic factors may shape leaf fungal communities by influencing plant traits (47). It was noteworthy that the impact of geographic location on aboveground compartment diversity was much greater than on belowground compartments (Table S4). Our results also demonstrated that diversity within plant compartments was lower than that outside, with interior compartments such as root endosphere being lower than rhizoplane and rhizosphere, and leaf endophytic being lower than leaf epiphytic (Fig. 1A and B). This indicated that the diversity of plant microbiomes gradually decreased from external to internal compartments. This may be due to the host's filtering effect (15, 48, 49) and possibly the plant's internal defense mechanisms inhibiting their proliferation (50). Particularly, the diversity of leaf-associated fungal communities is significantly higher than that in belowground compartments (Fig. 1A and B). Previous studies conducted in subtropical forest and mangrove ecosystems also found this result (51, 52). The higher heterogeneity of aboveground environments may lead to higher leaf-associated fungal diversity compared to roots with habitat homogeneity (51). In previous studies, fungal communities were more influenced by geographic location, which often led to different environmental factors shaping leaf-associated fungal communities (53).

### The fungal community composition between the aboveground and belowground compartments exhibited significant differences

Fungal community composition was shaped by compartment, geographic location, and plant identity (Fig. 3A). Previous research studies also supported the view that plant compartments were the main selective forces shaping microbiota, whether external or internal, and aboveground or belowground (15, 31). The influence of plant identity in the results was much smaller than compartment and geographic location (Fig. 3B), indicating a predominant role of biogeography in shaping plant-associated fungal communities (54). Fungal community composition exhibited significant differences between aboveground and belowground compartments (Fig. 2; Fig. S3), which corroborates findings from previous research studies on sorghum and hemp (55, 56). The aboveground and belowground compartments of plants represent completely different habitats (i.e., physical structure and chemical properties) resulting in selective recruitment of microbial communities (57, 58). Particularly, there was almost no exchange of fungi between

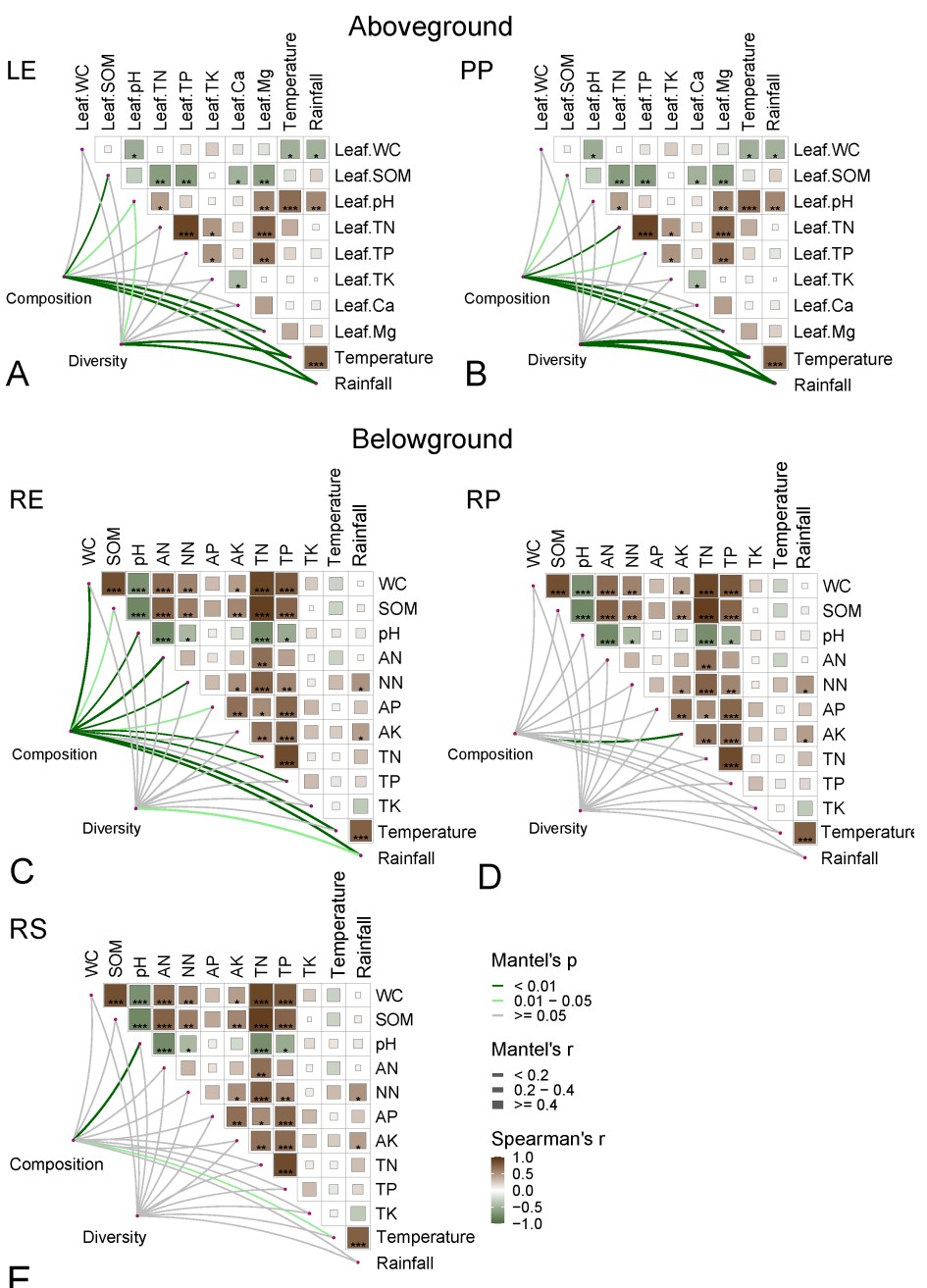

**FIG 4** Pairwise comparisons of the environmental factors are shown. Composition (OTU level) and diversity (observed OTU richness) of the community of different compartments of *D. pectinatum* and *V. mangachapoi* were related to each environmental factor by partial Mantel tests. Edge width corresponds to Mantel's *r* statistic for the corresponding distance correlation, and edge color denotes the statistical significance based on 999 permutations. Abbreviation: LE, leaf endophytic; PP, leaf epiphytic; RP, rhizoplane; RS, rhizosphere; and RE, root endosphere. Significance level: *$P < 0.05$; **$P < 0.01$; and ***$P < 0.001$.

aboveground and belowground compartments (Fig. S5). Previous studies have suggested that leaf-associated microorganisms may primarily come from the external environment, including the atmosphere and neighboring plants (59, 60). Additionally, geographic location had a greater impact on aboveground compartments than belowground compartments (Fig. 3B; Table S3), which is also reported in several other studies (61, 62). Aboveground and belowground compartments shape different fungal groups, each with distinct biotic and abiotic sensitivities. This results in variations in the response

levels of unique fungal groups to climatic environmental variables across different geographical locations (21).

## Driving factors for the fungal community were different between aboveground and belowground compartments

The general consensus is that leaf-associated microorganisms, exposed to the atmosphere, are more sensitive to environmental temperature, wind, and rainfall, compared to root-located, belowground microbes (50, 63, 64). Under these stressful conditions, leaf-associated fungi often rely on moisture and nutrients from the leaf or atmosphere, and, thus, these factors may influence fungal community compositions and diversity (65). In the results, temperature and rainfall were key factors driving the diversity of leaf-associated fungal communities in aboveground compartments (Fig. 4A, B, and 5A). It has been reported that climatic factors may shape leaf fungal communities by influencing plant traits (47). Suitable moisture and temperature conditions may promote active spore release, while rain splash may also contribute to fungal colonization (66–69). Temperature and rainfall have been shown to be key factors affecting the leaf fungi of rubber trees in tropical regions (14). Undoubtedly, climatic conditions are one of the major factors influencing leaf-associated microbial communities (70). On the contrary, root-associated fungal communities were more influenced by soil pH (Fig. 4C and E). Previous studies have shown that soil pH was one of the important factors affecting fungal communities (71). Soil pH can maintain the dynamic changes of microbial communities by affecting the development of host plant roots (72). In addition, rainfall, temperature, and leaf physicochemical properties (pH and Mg) were identified as important environmental drivers for the relative abundance of leaf-associated

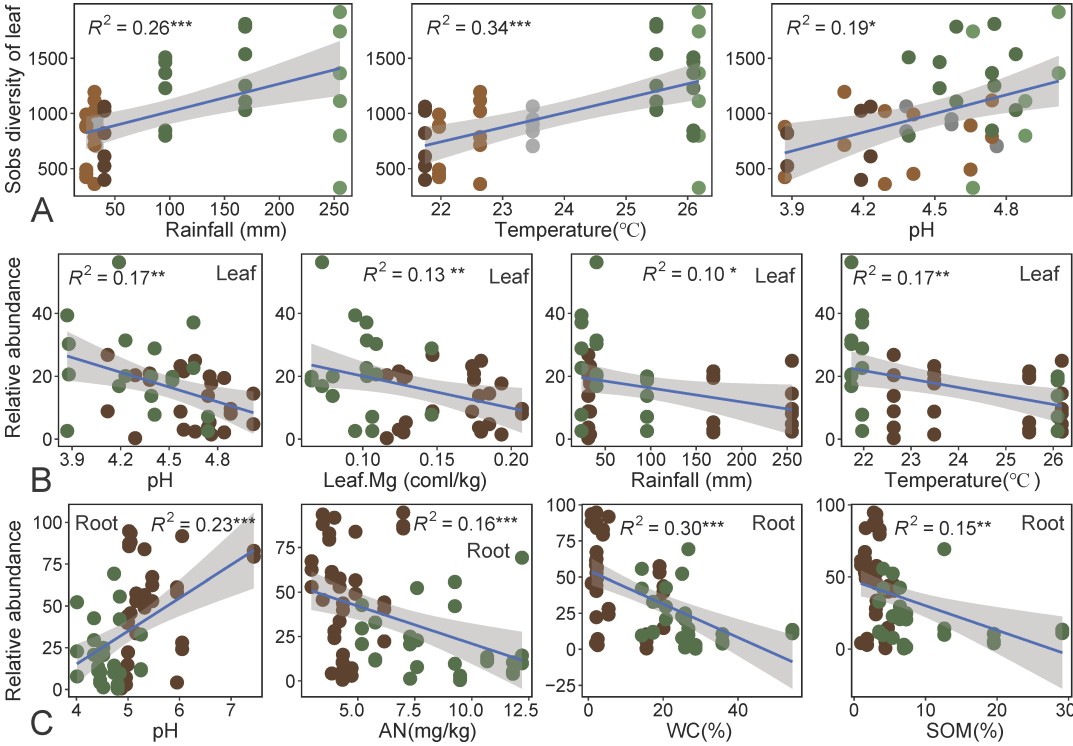

**FIG 5** The relationship among environmental factors, the diversity (Sobs index), and the relative abundance of *Eurotiomycetes* in leaf and *Agaricomycetes* in root in *D. pectinatum* and *V. mangachapoi*. (A) The relationship between environmental factors and Sobs index of the leaf. The light green solid circles represent samples of Bawang, and the dark green is Jianfeng. The light brown solid circles represent samples of Diaoluo, and the dark brown is Wuzhi. The dark gray solid circles represent samples of Wanning. (B) The relationship between environmental factors and the relative abundance of *Eurotiomycetes* in leaf. (C) The relationship between environmental factors and the relative abundance of *Agaricomycetes* in root. The dark green solid circles represent samples of *D. pectinatum*, and dark brown is *V. mangachapoi*. Significance level: \*$P < 0.05$; \*\*$P < 0.01$; \*\*\*$P < 0.001$.

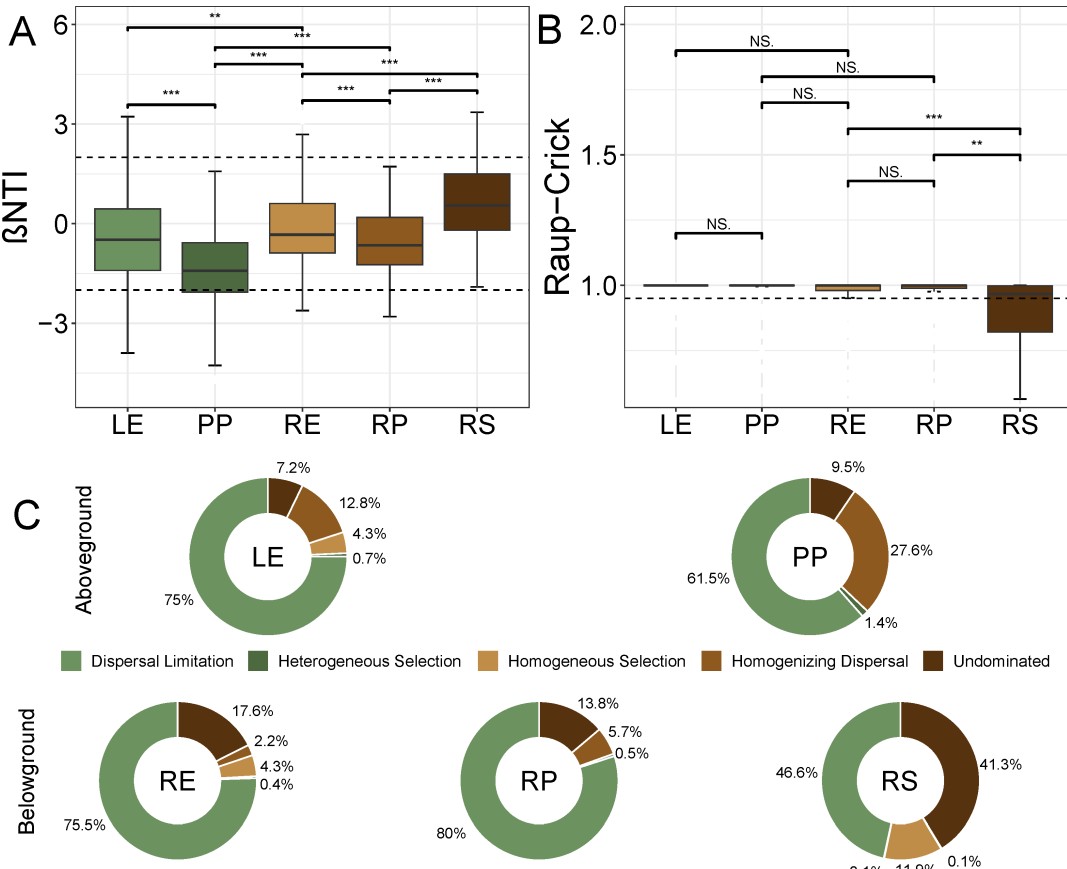

**FIG 6** Assembly processes of the fungal community in five compartments. (A and B) the boxplots of βNTI and Raup–Crick-based Bray–Curtis for all pairs of communities in five plant compartments. (C) the relative importance of different ecological processes in five plant compartments.Abbreviation: LE, leaf endophytic; PP, leaf epiphytic; RP, rhizoplane; RS, rhizosphere; and RE, root endosphere.

*Eurotiomycetes* (Fig. 5B). Climate change, particularly changes in temperature and rainfall, affects the growth and spread of fungal groups (73). Research indicates that temperature and humidity are critical determinants of the richness within the *Ascomycota* (74). Specifically, *Eurotiomycetes*, a significant subgroup within the Ascomycota phylum, are anticipated to be similarly affected by environmental perturbations. As a class of black yeasts, *Eurotiomycetes* contributes to equipping trees with defenses against the deleterious effects of ultraviolet radiation and in facilitating the decomposition of lignin. However, the looming threat of climate extremes in the future may induce fluctuations or even a decline in the populations of these fungi (75). Such a scenario could have profound and adverse consequences for the survival of endangered plants, exacerbating the challenges they face and heightening the urgency for conservation efforts. Additionally, *Agaricomycetes* are more likely to be enriched in *V. mangachapoi* than in *D. pectinatum*, with soil physicochemical properties (pH, AN, WC, and SOM) identified as key environmental drivers (Fig. 5C). *Agaricomycetes* are significantly correlated with soil pH, which may be related to its preference (76). Particularly, compared to belowground compartments, fungal diversity and composition in aboveground compartments seemed to be more strongly influenced by rainfall and temperature (Fig. 4). We find the same as a study on fungi in poplar at the continental scale (21). This may be because the open nature of the phyllosphere makes these fungi more susceptible to moisture loss than in soil environments (77). In detail, leaves are subjected to prolonged exposure in harsh environments, where resident microorganisms face various physical and chemical stresses such as UV radiation, rain splash, and drought. As a result, climatic factors such

as rainfall and temperature may shape the establishment of fungi in the phyllosphere (1, 50).

## Fungal community assemblages and functions in the aboveground compartment were different from those in the belowground

It is generally believed that deterministic factors strongly shape soil fungal community assemblages (78, 79). However, root microbiomes, as part of the soil microbiome, are thought to be governed by different assembly rules (5). Due to their small individual size and high community diversity, stochastic processes dominate the community assembly of microorganisms. Ecological stochasticity can be attributed to random processes, leading to random fluctuations in community structure in terms of species identity or functional traits (80). We observed stronger randomness in the belowground than in the aboveground compartments, and fungal colonization in the belowground compartments of plants seems to be more influenced by random variation (19, 54). Tropical rainforest soil may be typically characterized by low disturbance, and, therefore, be a relatively homogeneous and stable habitat, which results in a higher randomness in this compartment (81, 82). Further analysis revealed that dispersal limitation shaped fungal communities of aboveground and belowground compartments. The same results also indicated dispersal limitation is the primary process for rice phyllosphere fungal communities and soil fungal communities as well (43, 83). While fungi can produce abundant spores, this prolific spore production does not necessarily result in unrestricted dispersal (84). Consequently, fungi are significantly constrained by dispersal limitation (20, 27). FUNGuild analysis identified a considerable presence of ectomycorrhizal fungi in the belowground compartments of plants, with these groups exhibiting known dispersal limitations (43). *Agaricomycetes* were the dominant fungal class in the belowground compartments, commonly found as ectomycorrhizal fungi in forest soils (85). *Agaricomycetes* play a significant role as mycorrhizal fungi, highlighting their importance in the assembly of belowground fungal communities. Mycorrhizal fungi can provide timely nutrient replenishment to plants following extreme drought events, thereby facilitating rapid recovery of productivity (86). Preventing the decline of mycorrhizal fungal communities can help endangered tree species cope with future climate change. *Dothideomycetes*, known to infect most plants, are representatives of plant pathogenic fungi (87). A substantial proportion of plant pathogens and animal pathogens were predicted in the fungal communities of aboveground compartments, with *Dothideomycetes* being a significant member of this group. Most fungal plant pathogens exhibit clear host specificity and can only colonize under suitable environmental conditions, which may limit their spread during community assembly (88). Notably, the aboveground parts of trees may be more susceptible to pathogenic fungi. Under future climate change scenarios, the potential increase in pathogenic fungi could pose a greater threat to endangered plants (89, 90).

## Conclusions

We analyzed the diversity, composition, and community assembly patterns of fungal communities in endangered trees. In summary, fungal communities in leaves' and roots' compartments exhibited significant variation in their diversity between *D. pectinatum* and *V. mangachapoi*, and the factors driving these communities were distinct. Geographic location and plant species influenced aboveground fungal communities compared to those in belowground compartments. Soil pH emerged as the primary driver of belowground fungal communities, while rainfall and temperature predominated aboveground. Additionally, dispersal limitation mainly shaped fungal communities in aboveground and belowground compartments. Our results contribute to revealing the fungal diversity dynamics of tropical endangered plant species. We will need controlled laboratory experiments, followed by metagenomic sequencing for further research to understand the specific functional roles of microorganisms. This will allow us to better utilize their beneficial functions to enhance the growth adaptability of endangered trees.

## ACKNOWLEDGMENTS

The Hainan Province Science and Technology Special Fund (ZDYF2024SHFZ096), the Opening Foundation of Hainan Danzhou Tropical Agro-ecosystem National Observation and Research Station (RRI-KLOF202405), the National Natural Science Foundation of China (32271603), the Central Public-interest Scientific Institution Basal Research Fund (1630022022003), and the Earmarked Fund for Chinese Agricultural Research Systems (CARS-33-ZP3) funded this work.

## AUTHOR AFFILIATIONS

[1]Rubber Research Institute, Chinese Academy of Tropical Agricultural Sciences, Haikou City, Hainan Province, China

[2]College of Tropical Agriculture and Forestry, Hainan University, Haikou, China

[3]Hainan Danzhou Tropical Agro-ecosystem National Observation and Research Station, Danzhou City, Hainan Province, China

[4]College of Horticulture and Forestry, Huazhong Agricultural University, Wuhan, Hubei, China

[5]ECNU-Alberta Joint Lab for Biodiversity Study, Tiantong Forest Ecosystem National Observation and Research Station, School of Ecological and Environmental Sciences, East China Normal University, Shanghai, China

[6]College of Landscape Architecture and Forestry, Qingdao Agricultural University, Qingdao, Shandong, China

## AUTHOR ORCIDs

Kepeng Ji http://orcid.org/0009-0007-2428-3884
Yaqing Wei http://orcid.org/0000-0001-8157-4791
Yuwu Li http://orcid.org/0000-0003-3497-2898
Guoyu Lan http://orcid.org/0000-0003-4019-4252

## AUTHOR CONTRIBUTIONS

Kepeng Ji, Data curation, Formal analysis, Investigation, Methodology, Project administration, Resources, Software, Supervision, Validation, Visualization, Writing – original draft, Writing – review and editing | Yaqing Wei, Data curation, Formal analysis, Funding acquisition, Investigation, Methodology, Project administration, Resources, Software, Supervision | Xin Wang, Investigation, Software, Supervision | Yu Liu, Conceptualization, Methodology, Project administration, Resources, Supervision, Validation, Writing – review and editing | Rui Sun, Funding acquisition, Methodology, Resources, Supervision | Yuwu Li, Conceptualization, Funding acquisition, Supervision, Writing – review and editing | Guoyu Lan, Conceptualization, Data curation, Funding acquisition, Methodology, Project administration, Resources, Software, Supervision, Writing – review and editing

## DATA AVAILABILITY

The raw reads were deposited into the NCBI Sequence Read Archive (SRA) database (accession number: PRJNA1085516).

## ADDITIONAL FILES

The following material is available online.

### Supplemental Material

**Supplemental figures and table (Spectrum03092-24-s0001.docx).** Figures S1 to S6; Tables S1 to S5.

Open Peer Review

**PEER REVIEW HISTORY (review-history.pdf).** An accounting of the reviewer comments and feedback.

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
