## [Reviewer comments · Microbiology Spectrum]

Microbiology Spectrum

Distribution Patterns of Fungal Community Diversity in the Dominant Tree Species *Dacrydium pectinatum* and *Vatica mangachapoi* in Tropical Rainforests

Ji Kepeng, Yaqing Wei, wang xin, Guoyu Lan, Li yuwu, Yu Liu, and Rui Sun

Corresponding Author(s): Guoyu Lan, Rubber Research Institute, Chinese Academy of Tropical Agricultural Sciences; Hainan Danzhou Tropical Agro-ecosystem National Observation and Research Station

Review Timeline:

Submission Date:	November 28, 2024
Editorial Decision:	February 13, 2025
Revision Received:	February 28, 2025
Accepted:	March 24, 2025

Editor: Florian Freimoser

Reviewer(s): Disclosure of reviewer identity is with reference to reviewer comments included in decision letter(s). The following individuals involved in review of your submission have agreed to reveal their identity: Daolong Xu (Reviewer #1)

Transaction Report:

DOI: <https://doi.org/10.1128/spectrum.03092-24>

Re: Spectrum03092-24 (The Distinct Diversity Patterns of Fungal Communities in Aboveground and Belowground Compartments of Tropical Tree Species)

Dear Dr. Lan guoyu:

Thank you for the privilege of reviewing your work. Below you will find my comments, instructions from the Spectrum editorial office, and the reviewer comments. I will also send you, in a separate Email, a Word document with suggestions for your article from Reviewer 2.

Revision Guidelines

Sincerely,
Florian Freimoser
Editor
Microbiology Spectrum

Reviewer #1 (Comments for the Author):

The manuscript utilizes high-throughput sequencing technology to analyze the diversity and function of microbial communities in the rhizosphere and phyllosphere of two dominant tropical rainforest tree species, *Dacrydium pectinatum* and *Vatica mangachapoi*. This research is of great significance for the conservation of tropical rainforests. The paper is well-written with detailed methodologies.

1. It is recommended to change the title of the paper to "Distribution Patterns of Fungal Community Diversity in the Dominant

Tree Species *Dacrydium pectinatum* and *Vatica mangachapoi* in Tropical Rainforests." Since the analysis focuses on only these two tree species, reflecting this in the title would be appropriate.

2. (Lines 25-27): Propose to be modified as : To address this research gap, high-throughput sequencing technology was performed to investigate the diversity of fungal communities in leaves and roots compartments of *D. pectinatum* and *V. mangachapoi* from Hainan Island of China.

3. (Lines 25-26) In the abstract, "Aboveground compartments" should be revised to "leaves," and "belowground compartments" should be changed to "roots" to enhance reader comprehension.

4. In addition, for a research article no data were exhibited in the whole abstract, which was likely improper. Advise to supplement some key data results.

5. The references in the introduction section of the article need to be updated.

6. In the introduction, it is suggested to add one or two sentences briefly describing the basic characteristics of Hainan's tropical rainforest to highlight its importance in global biodiversity conservation.

7. (Lines 118): Propose to be modified as : Description of study area and site selection.

8. (Lines 131): Change the 2.2 subheading to "Sample collection and analysis".

9. (Lines 347-348):The abstract of the article mentions that soil pH mainly affects underground microorganisms, which needs to be discussed in this section to support your point.

10. (Lines 437-440): Propose to be modified as : In summary, fungal communities in leaves and roots compartments exhibited significant variation in their diversity between *Dacrydium pectinatum* and *Vatica mangachapoi*, and the factors driving these communities were distinct.

11. The vertical axis labels in Figures 5B and 5C should be "log(Abundance) of Eurotiomycetes" or "Agaricomycetes."

12. Please, be sure that all the references cited in the manuscript are also included in the reference list and vice versa with matching spellings and dates.

Reviewer #2 (Comments for the Author):

The manuscript by Ji and co-authors describes a detailed metabarcoding analysis of the fungal communities of two tropical trees, *Dacrydium pectinatum* and *Vatica mangachapoi*, in different above- and below-ground compartments. Samples were obtained from seven different locations (four for *V. mangachapoi*, three for *D. pectinatum*) and in total 105 samples were analysed. The study provides interesting insight in the fungal communities in different plant and soil compartments and compares also across sites. The authors also performed analyses to predict the most important factors affecting community structure and assembly.

Overall, it is an interesting and detailed analysis. A few general comments and suggestions for improvements are given below. Detailed corrections are directly suggested in the attached Word document (in track-change mode).

Lines142f: Please specify the buffered solution.

Line 148ff: Please provide more information about the collection of the soil samples.

Section 2.3: All words at the end of the line are split in a strange manner.

Line 192: Maybe provide some information and reference for the Sobs diversity index.

Section 3.2: Would it not be possible to provide more information on a lower taxonomic level? Ascomycota and Basidiomycota dominate virtually all compartments and niches anywhere. This is not informative. Maybe an analysis of the overlapping and not overlapping taxa would provide valuable information? Often people show Venn diagrams for this purpose. The authors may even analyze their data to see if a kind of core mycobiome could be defined.

Line 281: What is AN ?

Line 313: Is the diversity of bacteria also higher aboveground than in soil? This reviewer would have expected a higher diversity in soil.

Line 324: Does Figure 1C not suggest that plant identity does not affect the fungal community?

439 (first sentence of conclusion) is not complete.

Figures: It would be preferable to change the colors for the different sites at the two locations. The shades of green and brown are very difficult to distinguish; in particular in Figures 2 and 3-5. Maybe it would also be possible to add labels for the above- and belowground samples in each graph to make this distinction easier to recognize (for example a labelled bar).

Figure 2: It is not clear to this reviewer how the panels A and B are connected. There are a lot more Ascomycota in B than in A (based on percentage).

Microbiology Spectrum
Manuscript ID:03092-24

The manuscript utilizes high-throughput sequencing technology to analyze the diversity and function of microbial communities in the rhizosphere and phyllosphere of two dominant tropical rainforest tree species, *Dacrydium pectinatum* and *Vatica mangachapoi*. This research is of great significance for the conservation of tropical rainforests. The paper is well-written with detailed methodologies. However, there are some minor issues that should be addressed before publication.

1. It is recommended to change the title of the paper to "Distribution Patterns of Fungal Community Diversity in the Dominant Tree Species *Dacrydium pectinatum* and *Vatica mangachapoi* in Tropical Rainforests." Since the analysis focuses on only these two tree species, reflecting this in the title would be appropriate.
2. (Lines 25-27): Propose to be modified as : To address this research gap, high-throughput sequencing technology was performed to investigate the diversity of fungal communities in leaves and roots compartments of *D. pectinatum* and *V. mangachapoi* from Hainan Island of China.
3. (Lines 25-26) In the abstract, "Aboveground compartments" should be revised to "leaves," and "belowground compartments" should be changed to "roots" to enhance reader comprehension.
4. In addition, for a research article no data were exhibited in the whole abstract, which was likely improper. Advise to supplement some key data results.
5. The references in the introduction section of the article need to be updated.
6. In the introduction, it is suggested to add one or two sentences briefly describing the basic characteristics of Hainan's tropical rainforest to highlight its importance in global biodiversity conservation.
7. (Lines 118): Propose to be modified as : Description of study area and site selection.

8. (Lines 131): Change the 2.2 subheading to "Sample collection and analysis".
9. (Lines 347-348): The abstract of the article mentions that soil pH mainly affects underground microorganisms, which needs to be discussed in this section to support your point.
10. (Lines 437-440): Propose to be modified as: In summary, fungal communities in leaves and roots compartments exhibited significant variation in their diversity between *Dacrydium pectinatum* and *Vatica mangachapoi*, and the factors driving these communities were distinct.
11. The vertical axis labels in Figures 5B and 5C should be "log(Abundance) of Eurotiomycetes" or "Agaricomycetes."
12. Please, be sure that all the references cited in the manuscript are also included in the reference list and vice versa with matching spellings and dates.

**The Distinct Diversity Patterns of Fungal Communities in**
**Aboveground and Belowground Compartments of Tropical Tree**
**Species**

Kepeng Ji^{a,b†}, Yaqing Wei^{a,c†}, Xin Wang^{a,d}, Yu Liu^e, Guoyu Lan^{a,c*}, Yuwu Li^{f*}

a. Rubber Research Institute, Chinese Academy of Tropical Agricultural Sciences,
Haikou City, Hainan Province 571101, PR China

b. College of Tropical Agriculture and Forestry, Hainan University, Haikou 570228,
China

c. Hainan Danzhou Tropical Agro-ecosystem National Observation and Research
Station, Danzhou City, Hainan Province 571737, PR China

11 d. College of Horticulture and Forestry, Huazhong Agricultural University, Wuhan,
Hubei 430070, China

e. ECNU-Alberta Joint Lab for Biodiversity Study, Tiantong Forest Ecosystem
National Observation and Research Station, School of Ecological and
Environmental Sciences, East China Normal University, Shanghai 200241, China

f. College of Landscape Architecture and Forestry, Qingdao Agricultural University,
Qingdao, Shandong 266109, China

† These authors have contributed equally to this work

*Corresponding author: Guoyu Lan and Yuwu Li

E-mail: langyrri@163.com & liyuwu@qau.edu.cn

**Abstract:**

Plant microbial community are shaped by plant compartments, but the patterns of
fungal communities in aboveground and belowground compartments, and which
environmental factors can affect them, remain unknown. Here, we investigated fungal
communities in aboveground and belowground compartments of *Dacrydium*
*pectinatum* and *Vatica mangachapoi* from Hainan Island of China. Fungal
communities in aboveground and belowground compartments exhibited significant
differences. *Eurotiomycetes* and *Dothideomycetes* were predominantly found in
aboveground compartments, while *Agaricomycetes* dominated in belowground
compartments. Aboveground compartments had higher α -diversity. According to
Mantel test, soil pH mainly influenced the belowground compartments, while the

[revised manuscript text omitted]

142 kHz, and shaking the mixture at 200 rpm of four minutes. We repeated this procedure
three times. Finally, we filtered the solution using a vacuum filtration device and
stored the filter paper in a -80°C freezer (Bodenhausen et al., 2013; Ruiz-Perez et al.,
2016). The filtered leaf samples were washed with alcohol and then used for
sequencing of endophytic microorganisms. The rhizosphere soil was removed from
the root samples by hand, and then we used high-speed centrifugation to collect the
rhizoplane soil from the roots (Xiong et al., 2021).

**2.3 DNA extraction and PCR amplification**

We extracted fungal community genomic DNA from samples using the FastDNA® Sp
in Kit for Soil. All types of samples were operated according to the same kit. The spec
ific operation process was shown in Supporting Information Methods S2. We used PC
R primers ITS1F (5'-CTTGGTCATTTAGAGGAAGTAA-3') and ITS2R (5'-GCTGC
GTTCTTCATCGATGC-3') (Adams et al., 2013). The conditions for PCR amplificati
on included an initial denaturation step at 95°C for 3 minutes. This was followed by 3
5 cycles consisting of denaturation at 95°C for 30 seconds, annealing at 55°C for 30 s
econds, and extension at 72°C for 45 seconds. The process concluded with a final exte
nsion at 72°C for 10 minutes. Then the resultant PCR products were combined, and su
bsequent recovery was achieved through electrophoresis on a 2% agarose gel. Finally,

we purified the samples using the PCR Clean-Up Kit. Equimolar paired-end sequenci
161 ng (2×250 bp) was performed for the purified amplicons by the Illumina PE300 platf
orm in Shanghai Majorbio.

**2.4 Bioinformatics analysis**

The raw FASTQ files were demultiplexed, quality filtered, and merged using
fastp (version 0.19.6) (Magoč and Salzberg, 2011; Chen et al., 2018). The operation
was as follows: the terminal bases of reads with a quality value less than 20 were
filtered out, a window of 50 base pairs was set, and reads containing any "N" bases
were deleted. Double-end reads were merged into one sequence according to the
overlap (maximum overlap length does not exceed 10 base pairs; mismatch rate in the
overlap region of merged sequences does not exceed 0.2). Sequences that did not
meet this standard were removed. Samples were identified using the barcode and
primer located at the sequence ends, and sequence direction was adjusted accordingly.
The barcode had to match perfectly, while for the primer up to 2 mismatches were
allowed.

Operational taxonomic units (OTUs) were clustered with a similarity of no less
than 97% using UCHIME (version 7.2) (Edgar, 2013). The sequence count for
samples were rarefied to the minimum sample size with a uniform count of 30,101.
The taxonomy of sequences was identified by the RDP Classifier (version 2.2), with a
matching threshold set at 70% in UNITE fungal identification ITS database (Wang et
al., 2007; Nilsson et al., 2019; Lan et al., 2023). Please see Table S5 for the specific
OTU reports at different taxonomic levels.

**2.5 Statistic analysis**

We analyzed α -diversity index (OTU richness) by the *vegan* package (Dixon, 2003)
and the 'wilcox.test' of Wilcoxon rank-sum test was conducted in the *ggpubr* package
to compare results between two or more groups in R (version 4.2.3). The community
structure was analyzed by Principal Coordinate Analysis based on the Bray-Curtis
distance matrix and analyzed with PERMANOVA of the *vegan* package with 999
permutations (Zhang et al., 2022).

To investigate the potential driving factors of the Sobs diversity index, the
relative abundance of *Eurotiomycetes* in leaves and *Agaricomycetes* in roots, a
nonlinear regression model was used for evaluation by using ‘geom_smooth’ function
of R package ggplot2. The impact of soil properties, climate factors and leaf
properties on the fungal communities of different compartments was studied by using
‘mantel_tes’ function of the ggcors package (Huang et al., 2020). Fast expectation-
maximization microbial source tracking (FEAST) to reveal the source on fungal
communities was performed by FEAST package (Shenhav et al., 2019). FUNGuild
predicted fungal functions of the five compartments (Nguyen et al., 2016).

[revised manuscript text omitted]

community composition exhibited significant difference between aboveground and
belowground compartments (Figure 2 & Figure S3), which corroborate findings from
previous researches on sorghum and hemp (Wei et al., 2021; Sun et al., 2011). The
aboveground and belowground compartments of plants represent completely different
habitats (i.e., physical structure and chemical properties) resulted in selective
recruitment of microbial communities (Fitzpatrick et al., 2020; Trivedi et al., 2020).
Particularly, there was almost no exchange of fungi between aboveground and

belowground compartments (Figure S4). Previous studies have suggested that leaf-
associated microorganisms may primarily come from the external environment,
including the atmosphere and neighboring plants (Meyer et al., 2022; Lajoie &
Kembel, 2021). Additionally, geographic location had a greater impact on
aboveground compartments than belowground compartments (Figure 3B & Table S3),
which is also reported in several other studies (Shakya et al., 2013; Meiser et al.,
2014). Aboveground and belowground compartments shape different fungal groups,
each with distinct biotic and abiotic sensitivities. This results in variations in the
response levels of unique fungal groups to climatic environmental variables across
different geographical locations (Van Nuland et al., 2023).

**4.3 Driving factors for fungal community were different between aboveground** 345 **and belowground compartments.**

The general consensus is that leaf-associated microorganisms, exposed to the
atmosphere, are more sensitive to environmental temperature, wind and rainfall,
compared to root-located, belowground microbes (Sohrabi et al., 2023; Bashir et al.,
2022; Stone & Jackson, 2021). Under these stressful conditions, leaf-associated fungi
often rely on moisture and nutrients from leaf or atmosphere, and thus, these factors
may influence fungal community compositions and diversity (Gomes et al., 2018). In
the results, temperature and rainfall were key factors driving the diversity of leaf-
associated fungal communities in aboveground compartments (Figure 4AB & Figure
5A). It has been reported that climatic factors may shape leaf fungal communities by
influencing plant traits (Faticov et al., 2021). Suitable moisture and temperature
conditions may promote active spore release, while rain splash may also contribute to
fungal colonization (Kriel et al., 2000; Gigot et al., 2014; Martínez-Álvarez et al.,
2012; Hashizume et al., 2008). Temperature and rainfall have been shown to be key
factors affecting leaf fungi of rubber trees in tropical regions (Wei et al., 2022).
Undoubtedly, climatic conditions are one of the major factors influencing leaf-
associated microbial communities (Vacher et al., 2016). In addition, rainfall,
temperature, and leaf physicochemical properties (pH and Mg) were identified as

important environmental drivers for the relative abundance of leaf-associated
*Eurotiomycetes* (Figure 5B). Climate change, particularly changes in temperature and
rainfall, effect the growth and spread of fungal group (Andrew et al., 2016). Research
indicates that temperature and humidity are critical determinants of the richness
within the *Ascomycota* (Looby et al., 2018). Specifically, *Eurotiomycetes*, a
significant subgroup within the *Ascomycota* phylum, are anticipated to be similarly
affected by environmental perturbations. As a class of black yeasts, *Eurotiomycetes*
contributes to equip trees with defenses against the deleterious effects of ultraviolet
radiation and in facilitating the decomposition of lignin. However, the looming threat
of climate extremes in the future may induce fluctuations or even a decline in the
populations of these fungi (Dong et al., 2024). Such a scenario could have profound
and adverse consequences for the survival of endangered plants, exacerbating
challenges they face and heightening the urgency for conservation efforts.
Additionally, *Agaricomycetes* are more likely to be enriched in *V. mangachapoi* than
in *D. pectinatum.*, with soil physicochemical properties (pH, AN, WC, and SOM)
identified as key environmental drivers (Figure 5C). *Agaricomycetes* are significantly
correlated with soil pH, which may be related to its preference (Zhou et al., 2021).
Particularly, compared to belowground compartments, fungal diversity and
composition in aboveground compartments seemed to be more strongly influenced by
rainfall and temperature (Figure 4). We find the same as a study on fungi in poplar at
the continental scale (Van Nuland et al., 2023). This may be because the open nature
of the phyllosphere makes these fungi more susceptible to moisture loss than in soil
environments (Chen et al., 2021). In detail, leaves are subjected to prolonged
exposure in harsh environments, where resident microorganisms face various physical
and chemical stresses such as UV radiation, rain splash, and drought. As a result,
climatic factors such as rainfall and temperature may shape establishment of fungi in
phyllosphere (Müller et al., 2016; Sohrabi et al., 2023).

**4.4 Fungal community assemblages and functions in aboveground compartment**
**was different from that in belowground.**

It is generally believed that deterministic factors strongly shape soil fungal
community assemblages (Zheng et al., 2021; Guo et al., 2020). However, root
microbiomes, as part of the soil microbiome, are thought to be governed by different
assembly rules. (Attia et al., 2022). Due to their small individual size and high
community diversity, stochastic processes dominate community assembly of
microorganisms. Ecological stochasticity can be attributed to random processes,
leading to random fluctuations in community structure in terms of species identity or
functional traits (Zhou & Ning., 2017). We observed stronger randomness in the
belowground than in the aboveground compartments, and fungal colonization in the
belowground compartments of plants seems to be more influenced by random
variation (Coleman-Derr et al., 2016; Zhang et al., 2018). Tropical rainforest soil may
be typically characterized by low disturbance, and therefore be a relatively
homogeneous and stable habitat, which results in a higher randomness in this
compartment (Wang et al., 2013; Jiao et al., 2020). Further analysis revealed that
dispersal limitation shaped fungal communities of aboveground and belowground
compartments. Same results also indicated dispersal limitation being the primary
process for rice phyllosphere fungal communities and soil fungal communities as well
(Yin et al., 2023; Osburn et al., 2021). While fungi can produce abundant spores, this
prolific spore production does not necessarily result in unrestricted dispersal (Peay et
al., 2012). Consequently, fungi are significantly constrained by dispersal limitation
(Talbot et al., 2014; Taylor et al., 2006). FUNGuild analysis identified a considerable
presence of ectomycorrhizal fungi in the belowground compartments of plants, with
these groups exhibiting known dispersal limitations (Osburn et al., 2021).
*Agaricomycetes* were the dominant fungal class in the belowground compartments,
commonly found as ectomycorrhizal fungi in forest soils (Hout et al., 2024).
*Agaricomycetes* play a significant role as mycorrhizal fungi, highlighting their
importance in the assembly of belowground fungal communities. Mycorrhizal fungi
can provide timely nutrient replenishment to plants following extreme drought events,
thereby facilitating rapid recovery of productivity (Liu et al., 2022). Preventing the
decline of mycorrhizal fungal communities can help endangered tree species cope

with future climate change. *Dothideomycetes*, known to infect most plants, are
representatives of plant pathogenic fungi (Haridas et al., 2020). A substantial
proportion of plant pathogens and animal pathogens were predicted in the fungal
communities of aboveground compartments, with *Dothideomycetes* being a
significant member of this group. Most fungal plant pathogens exhibit clear host
specificity and can only colonize under suitable environmental conditions, which may
limit their spread during community assembly (Inoue et al., 2023). Notably, the
aboveground parts of trees may be more susceptible to pathogenic fungi. Under future
climate change scenarios, the potential increase in pathogenic fungi could pose a
greater threat to endangered plants (Yang et al., 2023; Chaloner et al., 2021).

**5. Conclusions**

Analyzing the diversity, composition and community assembly patterns of fungal
communities of endangered trees. Our results indicated that fungal communities in
aboveground and belowground compartments exhibited significant differences, and
the factors driving these communities were distinct. Geographic location and plant
species influenced aboveground fungal communities compared to those in
belowground compartments. Soil pH emerged as the primary driver belowground
fungal communities, while rainfall and temperature predominated aboveground.
Additionally, dispersal limitation mainly shaped fungal communities in aboveground
and belowground compartments. Our results contribute to reveal fungal diversity
dynamics of tropical endangered plant species. We will need controlled laboratory
experiments, followed by metagenomic sequencing for further research to understand
the specific functional roles of microorganisms. This will allow us to better utilize
their beneficial functions to enhance the growth adaptability of endangered trees.

**Author Contributions**

Guoyu Lan and Yuwu Li provided study ideas; Kepeng Ji, Yaqing Wei. and Xing
Wang conducted experiments; Kepeng Ji proceeded formal analysis and wrote final

draft. Guoyu Lan accomplished final edit. Both authors approved the reported work.

**Acknowledgements**

The Hainan Province Science and Technology Special Fund (ZDYF2024SHFZ096),
the Opening Foundation of Hainan Danzhou Tropical Agro-ecosystem National
Observation and Research Station (RRI-KLOF202405), the National Natural Science
Foundation of China (32271603), and the Earmarked Fund for Chinese Agricultural
Research Systems (CARS-33-ZP3) funded this work.

**Data Availability**

The raw reads were deposited into the NCBI Sequence Read Archive (SRA) database
(Accession Number: PRJNA1085516).

**Reference**

- Adams, R. I., Miletto, M., Taylor, J. W., Bruns, T. D. (2013). Dispersal in microbes: fungi in indoor
 air are dominated by outdoor air and show dispersal limitation at short distances. *The ISME*
 *journal*, 7(7), 1262-1273. <https://doi.org/10.1038/ismej.2013.28>.
- Attia, S., Russel, J., Mortensen, M. S., Madsen, J. S., Sørensen, S. J. (2022). Unexpected diversity
 among small-scale sample replicates of defined plant root compartments. *The ISME journal*,
 16(4), 997-1003. <https://doi.org/10.1038/s41396-021-01094-7>.
- Andrew, C., Heegaard, E., Halvorsen, R., Martinez-Peña, F., Egli, S., Kirk, P. M., ... & Kauserud,
 H. (2016). Climate impacts on fungal community and trait dynamics. *Fungal Ecology*, 22, 17-
 25. <https://doi.org/10.1016/j.funeco.2016.03.005>.
- Bashir, I., War, A. F., Rafiq, I., Reshi, Z. A., Rashid, I., Shouche, Y. S. (2022). Phyllosphere micr
 obiome: Diversity and functions. *Microbiological Research*, 254, 126888. <https://doi.org/10.1016/j.micres.2021.126888>. <https://doi.org/10.1007/s11104-013-1778-x>.
- Bodenhausen, N., Horton, M. W., Bergelson, J. (2013). Bacterial communities associated with the
 leaves and the roots of *Arabidopsis thaliana*. *PloS one*, 8(2), e56329. <https://doi.org/10.1371/journal.pone.0056329>.
- Berhongaray, G., Janssens, I. A., King, J. S., and Ceulemans, R. (2013). Fine root biomass and tur
 nlover of two fast-growing poplar genotypes in a short-rotation coppice culture. *Plant Soil*, 37
 3(1-2): 269-283.
- Chaloner, T. M., Gurr, S. J., Bebbler, D. P. (2021). Plant pathogen infection risk tracks global crop
 yields under climate change. *Nature Climate Change*, 11(8), 710-715. <https://doi.org/10.1038/s41558-021-01104-8>.
- Chen, S., Zhou, Y., Chen, Y., & Gu, J. (2018). fastp: an ultra-fast all-in-one FASTQ preprocesso
 r. *Bioinformatics*, 34(17), i884-i890. <https://doi.org/10.1093/bioinformatics/bty560>.
- Chen, Q., Hu, H., Yan, Z., Li, C., Nguyen, B. A. T., Zhu, Y., He, J. (2021). Precipitation increases t
 he abundance of fungal plant pathogens in *Eucalyptus* phyllosphere. *Environmental Microbio*
 *logy*, 23(12), 7688-7700. <https://doi.org/10.1111/1462-2920.15728>.
- Coleman-Derr, D., Desgarences, D., Fonseca-Garcia, C., Gross, S., Clingenpeel, S., Woyke, T., No
 rth, G., Visel, A., Partida-Martinez, L.P., Tringe, S.G. (2016). Plant compartment and biogeog
 raphy affect microbiome composition in cultivated and native *Agave* species. *New Phytologis*
 *t*, 209(2), 798-811. <https://doi.org/10.1111/nph.13697>.
- Cregger, M. A., Veach, A. M., Yang, Z. K., Crouch, M. J., Vilgalys, R., Tuskan, G. A., Schadt, C.
 494 W. (2018). The *Populus* holobiont: dissecting the effects of plant niches and genotype on the
 495 microbiome. *Microbiome*, 6(1), 1-14. <https://doi.org/10.1186/s40168-018-0413-8>.
- de Vries, F. T., Griffiths, R. I., Knight, C. G., Nicolitch, O., Williams, A. (2020). Harnessing rhizo
 sphere microbiomes for drought-resilient crop production. *Science*, 368(6488), 270-274. <https://doi.org/10.1126/science.aaz5192>.
- Dong, L., Li, M. X., Li, S., Yue, L. X., Ali, M., Han, J. R., Lian, W. H., Hu C. J., Lin, Z. L., Shi, G.
 Y., Wang, P. D., Gao, S. M., Lian, Z. H., She, T. T., Wei, Q. C., Deng, Q. Q., Hu, Q., Xiong,
 501 J. L., Liu, Y. H., Li, W. J. (2024). Aridity drives the variability of desert soil microbiomes ac
 ross north-western China. *Science of the Total Environment*, 907, 168048. <https://doi.org/10.1016/j.scitotenv.2023.168048>.
- Dixon, P., 2003. VEGAN, a package of R functions for community ecology. *J. Veg. Sci.* 14, 927–9

30. <https://doi.org/10.1111/j.1654-1103.2003.tb02228.x>.

Edgar, R.C. (2013). UPARSE: highly accurate OTU sequences from microbial amplicon reads. *Nature Methods*, 10: 996-1008. <https://doi.org/10.1038/nmeth.2604>.

Edwards, J., Johnson, C., Santos-Medellín, C., Lurie, E., Podishetty, N. K., Bhatnagar, S., Eisen, J.

510 A., Sundaresan, V. (2015). Structure, variation, and assembly of the root-associated microbiomes of rice. *Proceedings of the National Academy of Sciences*, 112(8), E911-E920. <https://doi.org/10.1073/pnas.1414592112>.

Faticov, M., Abdelfattah, A., Roslin, T., Vacher, C., Hambäck, P., Blanchet, F. G., Lindahl, B.D., T

ack, A. J. (2021). Climate warming dominates over plant genotype in shaping the seasonal trajectory of foliar fungal communities on oak. *New Phytologist*, 231(5), 1770-1783. <https://doi.org/10.1111/nph.17434>.

Fitzpatrick, C. R., Salas-González, I., Conway, J. M., Finkel, O. M., Gilbert, S., Russ, D., Teixeira,

P.J.P.L., Dangl, J. L. (2020). The plant microbiome: from ecology to reductionism and beyond. *Annual review of microbiology*, 74, 81-100. <https://doi.org/10.1146/annurev-micro-022620-014327>.

Fitzpatrick, C. R., Copeland, J., Wang, P. W., Guttman, D. S., Kotanen, P. M., Johnson, M. T. (201

8). Assembly and ecological function of the root microbiome across angiosperm plant species. *Proceedings of the National Academy of Sciences*, 115(6), E1157-E1165. <https://doi.org/10.1073/pnas.1717617115>.

Gong T, Xin X. F. (2022). Phyllosphere microbiota: Community dynamics and its interaction with

plant hosts. *Journal of Integrative Plant Biology*, 63(2): 297-304. <https://doi.org/10.1111/jipb.13060>.

Gigot, C., de Vallavieille-Pope, C., Huber, L., SaintJean, S. (2014). Using virtual 3-D plant architecture to assess fungal pathogen splash dispersal in heterogeneous canopies: a case study with

cultivar mixtures and a non-specialized disease causal agent. *Annals of Botany*, (4), 863-875.

<https://doi.org/10.1093/aob/mcu098>.

Gomes, T., Pereira, J. A., Benhadi, J., Lino-Neto, T., Baptista, P. (2018). Endophytic and epiphytic

phyllosphere fungal communities are shaped by different environmental factors in a Mediterranean ecosystem, *Microbial ecology*, 76, 668-679. <https://doi.org/10.1007/s00248-018-1161-9>.

Guo, J., Ling, N., Chen, Z., Xue, C., Li, L., Liu, L., Gao, L., Wang, M., Ruan, J., Guo, S., Vandenkoornhuyse, P., Shen, Q. (2020). Soil fungal assemblage complexity is dependent on soil fertility and dominated by deterministic processes. *New Phytologist*, 226(1), 232-243. <https://doi.org/10.1111/nph.16345>.

Hanson, C.A., Fuhrman, J.A., Horner-Devine, M.C., Martiny, J.B. (2012). Beyond biogeographic

patterns: processes shaping the microbial landscape. *Nat. Rev. Microbiol.* 10, 497–506. <https://doi.org/10.1038/nrmicro2795>.

Hashizume, Y., Sahashi, N., Fukuda, K. (2008). The influence of altitude on endophytic mycobiota in *Quercus acuta* leaves collected in two areas 1000 km apart. *Forest Pathology*, 38(3), 218-

226. <https://doi.org/10.1111/j.1439-0329.2008.00547.x>.

Hassani, M.A., Durán, P., Hacquard, S. (2018). Microbial interactions within the plant holobiont, *Microbiome*, 6, 1-17. <https://doi.org/10.1186/s40168-018-0445-0>.

Haridas, S., Albert, R., Binder, M., Bloem, J., LaButti, K., Salamov, A., et al. (2020). 101 Dothideomycetes genomes: a test case for predicting lifestyles and emergence of pathogens. *Studies in*

n mycology, 96(1), 141-153. <https://doi.org/10.1016/j.simyco.2020.01.003>.

Howe, A., Stopnisek, N., Dooley, S. K., Yang, F., Grady, K. L., Shade, A. (2023). Seasonal activit
ies of the phyllosphere microbiome of perennial crops. *Nature Communications*, 14(1), 1039.
<https://doi.org/10.1038/s41467-023-36515-y>.

Huang, H.Y., Zhou, L., Chen, J., Wei, T.Y. (2020). ggcOR: Extended tools for correlation analysis a
nd visualization. R package version 0.9.7.

Hou, J., McCormack, M. L., Reich, P. B., Sun, T., Phillips, R. P., Lambers, H., Chen, H. Y. H., Din
557 g, Y. Y., Comas, L. H., Valverde-Barrantes, O. J., Solly, E. F., Freschet, G. T. (2024). Linking
fine root lifespan to root chemical and morphological traits—A global analysis. *Proceedings
of the National Academy of Sciences*, 121(16), e2320623121. [https://doi.org/10.1073/pnas.23
20623121](https://doi.org/10.1073/pnas.23

560 20623121).

Inoue, Y., Phuong Vy, T. T., Singkaravanit-Ogawa, S., Zhang, R., Yamada, K., Ogawa, T., Ishizuk
562 J., Narusaka, Y., Takano, Y. (2023). Selective deployment of virulence effectors correlates wit
563 h host specificity in a fungal plant pathogen. *New Phytologist*, 238(4), 1578-1592. [https://doi.
org/10.1111/nph.18790](https://doi.

org/10.1111/nph.18790).

Jiao, S., Yang, Y., Xu, Y., Zhang, J., Lu, Y. (2020). Balance between community assembly process
es mediates species coexistence in agricultural soil microbiomes across eastern China. *The IS
ME Journal*, 14(1), 202-216. <https://doi.org/10.1038/s41396-019-0522-9>.

Kembel, S. W., O'Connor, T. K., Arnold, H. K., Hubbell, S. P., Wright, S. J., Green, J. L. (2014). R
elationships between phyllosphere bacterial communities and plant functional traits in a neotr
opical forest. *Proceedings of the National Academy of Sciences*, 111(38), 13715-13720. [http
571 s://doi.org/10.1073/pnas.1216057111](http

571 s://doi.org/10.1073/pnas.1216057111).

Kriel, W. M., Swart, W. J., Crous, P. W. (2000). Foliar endophytes and their interactions with host
plants, with specific reference to the gymnospermae. *Advances in Botanical Research*, 33, 1-
34. [https://doi.org/10.1016/S0065-2296\(00\)33040-3](https://doi.org/10.1016/S0065-2296(00)33040-3).

Lajoie, G., Kembel, S. W. (2021). Host neighborhood shapes bacterial community assembly and s
pecialization on tree species across a latitudinal gradient. *Ecological Monographs*, 91(2), e01
443. <https://www.jstor.org/stable/27081077>.

Lan, G., Wei, Y., Li, Y., Wu, Z. (2023). Diversity and assembly of root-associated microbiomes of
rubber trees. *Frontiers in Plant Science*, 14, 1136418. [https://doi.org/10.3389/fpls.2023.11364
18](https://doi.org/10.3389/fpls.2023.11364

18).

Li, M., Hong, L., Ye, W., Wang, Z., Shen, H. (2022). Phyllosphere bacterial and fungal communiti
es vary with host species identity, plant traits and seasonality in a subtropical forest. *Environ
mental microbiome*, 17(1), 1-13. <https://doi.org/10.1186/s40793-022-00423-3>.

Liu, H., Macdonald, C. A., Cook, J., Anderson, I. C., Singh, B. K. (2019). An ecological loop: host
microbiomes across multitrophic interactions. *Trends in ecology & evolution*, 34(12), 1118-1
130. <https://doi.org/10.1016/j.tree.2019.07.011>.

Liu, S., García-Palacios, P., Tedersoo, L., Guirado, E., van der Heijden, M. G., Wagg, C., Chen,
D., Wang, Q., Wang, J., K. Singh, B., Delgado-Baquerizo, M. (2022). Phylotype diversity wit
hin soil fungal functional groups drives ecosystem stability. *Nature Ecology & Evolution*, 6
(7), 900-909. <https://doi.org/10.1038/s41559-022-01756-5>.

Looby, C. I., & Treseder, K. K. (2018). Shifts in soil fungi and extracellular enzyme activity with
simulated climate change in a tropical montane cloud forest. *Soil Biology and Biochemistr
y*, 117, 87-96. <https://doi.org/10.1016/j.soilbio.2017.11.014>.

Magoč, T., & Salzberg, S. L. (2011). FLASH: fast length adjustment of short reads to improve gen
ome assemblies. *Bioinformatics*, 27(21), 2957-2963. <https://doi.org/10.1093/bioinformatics/btr507>
r507

Martínez-Álvarez, P., Rodríguez-Ceinós, S., Martín-García, J., Diez, J. J. (2012). Monitoring endo
phyte populations in pine plantations and native oak forests in Northern Spain. *Forest System*
599 s, 21(3), 373-382. <https://doi.org/10.5424/fs/2012213-02254>.

Meiser, A., Bálint, M., Schmitt, I. (2014). Meta-analysis of deep-sequenced fungal communities in
dicates limited taxon sharing between studies and the presence of biogeographic patterns. *New*
*Phytologist*, 201(2), 623-635. <https://doi.org/10.1111/nph.12532>.

Mendes, L.W., Kuramae, E.E., Navarrete, A.A., van Veen, J.A., Tsai, S.M. (2014). Taxonomical an
604 d functional microbial community selection in soybean rhizosphere. *The ISME journal*, 8(8),
1577-1587. <https://doi.org/10.1038/ismej.2014.17>.

Meyer, K. M., Porch, R., Muscettola, I. E., Vasconcelos, A. L. S., Sherman, J. K., Metcalf, C. J. E.,
Lindow, S.E., Koskella, B. (2022). Plant neighborhood shapes diversity and reduces interspe
cific variation of the phyllosphere microbiome. *The ISME journal*, 16(5), 1376-1387. <https://doi.org/10.1038/s41396-021-01184-6>.

Morris, C. E. (2001). Phyllosphere. *eLS*. <https://doi.org/10.1038/npg.els.0000400>.

Mueller, E. A., Wisnoski, N. I., Peralta, A. L., Lennon, J. T. (2020). Microbial rescue effects: how
microbiomes can save hosts from extinction. *Functional Ecology*, 34(10), 2055-2064. <https://doi.org/10.1111/1365-2435.13493>.

Müller, D.B., Vogel, C., Bai, Y., Vorholt, J.A. (2016), The plant microbiota: Systems-level insights
and perspectives. *Annu Rev Genet*, 50: 211-234. <https://doi.org/10.1146/annurev-genet-120215-034952>.

Ning, D., Yuan, M., Wu, L., Zhang, Y., Guo, X., Zhou, X., Yang, Y., Arkin, A., Firestone, M., Zho
u, J. (2020). A quantitative framework reveals ecological drivers of grassland microbial com
munity assembly in response to warming, *Nature communications*, 11(1), 4717. <https://doi.org/10.1038/s41467-020-18560-z>.

Nilsson, R. H., Larsson, K. H., Taylor, A. F. S., Bengtsson-Palme, J., Jeppesen, T. S., Schigel, D.,
... & Abarenkov, K. (2019). The UNITE database for molecular identification of fungi: handl
ing dark taxa and parallel taxonomic classifications. *Nucleic acids research*, 47(D1), D259-D
264. <https://doi.org/10.1093/nar/gky1022>.

Nguyen, N. H., Song, Z., Bates, S. T., Branco, S., Tedersoo, L., Menke, J., et al. (2016). FUNGuil
626 d: an open annotation tool for parsing fungal community datasets by ecological guild. *Fungal*
*Ecol.* 20, 241–248. <https://doi.org/10.1016/j.funeco.2015.06.006>.

Osburn, E.D., Aylward, F.O. and Barrett, J.E. (2021). Historical land use has long-term effects on
microbial community assembly processes in forest soils. *ISME Communications*, 1(1), 48. <https://doi.org/10.1038/s43705-021-00051-x>.

Peay, K. G., Schubert, M. G., Nguyen, N. H., & Bruns, T. D. (2012). Measuring ectomycorrhizal f
ungal dispersal: macroecological patterns driven by microscopic propagules. *Molecular Ecol*
*ogy*, 21(16), 4122-4136. <https://doi.org/10.1111/j.1365-294X.2012.05666.x>

Rosado, B. H., Almeida, L. C., Alves, L. F., Lambais, M. R., Oliveira, R. S. (2018). The importanc
e of phyllosphere on plant functional ecology: a phyllo trait manifesto. *New Phytologist*, 219
(4), 1145-1149. <https://doi.org/10.1111/nph.15235>.

Ruiz-Perez, C. A., Restrepo, S., Zambrano, M. M. (2016). Microbial and functional diversity withi

n the phyllosphere of Espeletia species in an Andean high-mountain ecosystem. Applied and
Environmental Microbiology, 82(6), 1807-1817. <https://doi.org/10.1128/AEM.02781-15>.

Santamaría, J., Bayman, P. (2005). Fungal epiphytes and endophytes of coffee leaves (*Coffea arabi*
*ca*). Microbial ecology, 50, 1-8. <https://www.jstor.org/stable/25153220>.

Shakya, M., Gottel, N., Castro, H., Yang, Z. K., Gunter, L., Labbé, J., Muchero, W., Bonito, G., Vi
lgalys, R., Tuskan, G., Podar, M., Schadt, C. W. (2013). A multifactor analysis of fungal and
bacterial community structure in the root microbiome of mature *Populus deltoides* trees. PloS
one, 8(10), e76382. <https://doi.org/10.1371/journal.pone.0076382>.

Shenhav, L., Thompson, M., Joseph, T. A., Briscoe, L., Furman, O., Bogumil, D., Mizrahi, I., Pe'e
r I., Halperin, E. (2019). FEAST: fast expectation-maximization for microbial source trackin
648 g. Nature methods, 16(7), 627-632. <https://doi.org/10.1038/s41592-019-0431-x>.

Sohrabi, R., Paasch, B. C., Liber, J. A., He, S. Y. (2023). Phyllosphere microbiome. Annual Revie
w of Plant Biology, 74, 539-568. <https://doi.org/10.1146/annurev-arplant-102820-032704>.

Stegen, J. C., Lin, X., Fredrickson, J. K., Chen, X., Kennedy, D. W., Murray, C. J., Rockhold, M.
652 J., Konopka, A. (2013). Quantifying community assembly processes and identifying features
that impose them. The ISME journal, 7(11), 2069-2079. <https://doi.org/10.1038/ismej.2013.9>
3.

Stone, B.W.G., Jackson, C. R. (2021). Seasonal patterns contribute more towards phyllosphere ba
cterial community structure than short-term perturbations. Microbial ecology, 81, 146-156. ht
tps://doi.org/10.1007/s00248-020-01564-z.

Sun, A., Jiao, X.Y., Chen, Q., Wu, A.L., Zheng, Y., Lin, Y.X., He, J.Z., Hu, H. W. (2021). Microbia
659 l communities in crop phyllosphere and root endosphere are more resistant than soil microbio
ta to fertilization. Soil Biology and Biochemistry, 153, 108113. <https://doi.org/10.1016/j.soilb>
io.2020.108113.

Steidinger, B. S., Bhatnagar, J. M., Vilgalys, R., Taylor, J. W., Qin, C., Zhu, K., ... & Peay, K. G.
(2020). Ectomycorrhizal fungal diversity predicted to substantially decline due to climate cha
nges in North American Pinaceae forests. Journal of biogeography, 47(3), 772-782. <https://do>
i.org/10.1111/jbi.13802.

Talbot, J.M., Bruns, T.D., Taylor, J.W., Smith, D.P., Branco, S., Glassman, S.I., Erlandson, S., Vilg
alys, R., Liao, H.L., Smith, M.E., Peay, K.G. (2014). Endemism and functional convergence
across the North American soil mycobiome. Proceedings of the National Academy of Science
669 s, 111(17), 6341-6346. <https://doi.org/10.1073/pnas.1402584111>.

Taylor, J. W., Turner, E., Townsend, J. P., Dettman, J. R., Jacobson, D. (2006). Eukaryotic microbe
671 s, species recognition and the geographic limits of species: examples from the kingdom Fung
i. Philosophical Transactions of the Royal Society B: Biological Sciences, 361(1475), 1947-1
963. <https://doi.org/10.1098/rstb.2006.1923>.

Trivedi, P., Leach, J. E., Tringe, S. G., Sa, T., Singh, B. K. (2020). Plant–microbiome interactions:
from community assembly to plant health. Nature reviews microbiology, 18(11), 607-621. htt
ps://doi.org/10.1038/s41579-020-0412-1.

Vacher, C., Hampe, A., Porté, A. J., Sauer, U., Compant, S., Morris, C. E. (2016). The phyllospher
e: microbial jungle at the plant–climate interface. Annual review of ecology, evolution, and s
ystematics, 47, 1-24. <https://doi.org/10.1146/annurev-ecolsys-121415-032238>.

Vacher, C., Cordier, T., Vallance, J. (2016). Phyllosphere fungal communities differentiate more t
horoughly than bacterial communities along an elevation gradient. Microbial ecology, 72, 1-3.

<https://doi.org/10.1007/s00248-016-0742-8>.

Van Nuland, M. E., Daws, S. C., Bailey, J. K., Schweitzer, J. A., Busby, P. E., Peay, K. G. (2023).
Above-and belowground fungal biodiversity of *Populus* trees on a continental scale. *Nature*
*Microbiology*, 8(12), 2406-2419. <https://doi.org/10.1038/s41564-023-01514-8>.

Vandenkoornhuysen, P., Quaiser, A., Duhamel, M., Le Van, A., Dufresne, A. (2015). The importanc
e of the microbiome of the plant holobiont. *New Phytologist*, 206(4), 1196-1206. [https://doi.o](https://doi.org/10.1111/nph.13312)
[rg/10.1111/nph.13312](https://doi.org/10.1111/nph.13312).

van der Linde, S., Suz, L. M., Orme, C. D. L., Cox, F., Andreae, H., Asi, E., ... & Bidartondo, M. I.
(2018). Environment and host as large-scale controls of ectomycorrhizal fungi. *Nature*, 558
(7709), 243-248. <https://doi.org/10.1038/s41586-018-0189-9>.

Wang, B., Zhang, W. (2002). The groups and features of tropical forest vegetation of Hainan Islan
693 d. *Guihaia*, (2):107-115.

Wang, J., Shen, J. I., Wu, Y., Tu, C., Soininen, J., Stegen, J. C., He, J., Liu, X., Zhang, L., Zhang,
E. (2013). Phylogenetic beta diversity in bacterial assemblages across ecosystems: determinis
tic versus stochastic processes. *The ISME journal*, 7(7), 1310-1321. [https://doi.org/10.1038/ism](https://doi.org/10.1038/ismej.2013.30)
[ej.2013.30](https://doi.org/10.1038/ismej.2013.30).

Wang, Q., Garrity, G.M., Tiedje, J.M., and Cole, J.R. (2007). Naive Bayesian classifier for rapid as
signment of rRNA sequences into the new bacterial taxonomy. *Appl Environ Microbiol*, 73: 5
261-5267. <https://doi.org/10.1128/AEM.00062-07>

Wei, G., Ning, K., Zhang, G., Yu, H., Yang, S., Dai, F., Dong, L., Chen, S. (2021). Compartment ni
che shapes the assembly and network of *Cannabis sativa*-associated microbiome. *Frontiers in*
*Microbiology*, 12, 714993. <https://doi.org/10.3389/fmicb.2021.714993>.

Wei, Y., Lan, G., Wu, Z., Chen, B., Quan, F., Li, M., Su, S., Du, H. (2022). Phyllosphere fungal co
mmunities of rubber trees exhibited biogeographical patterns, but not bacteria. *Environmental*
*Microbiology*, 24(8), 3777-3790. <https://doi.org/10.1111/1462-2920.15894>.

Xiong, C., Zhu, Y. G., Wang, J. T., Singh, B., Han, L. L., Shen, J. P., Li, P.P., Wang, G.B., Wu, C.
F., Ge, A.H., Zhang, L.M., He, J. Z. (2021). Host selection shapes crop microbiome assembly
and network complexity. *New Phytologist*, 229(2), 1091-1104. [https://doi.org/10.1111/nph.1](https://doi.org/10.1111/nph.16890)
[6890](https://doi.org/10.1111/nph.16890).

Yang, H., Yang, Z., Wang, Q. C., Wang, Y. L., Hu, H. W., He, J. Z., Zheng, Y., Yang, Y. (2022). Co
mpartment and plant identity shape tree mycobiome in a subtropical forest. *Microbiology Spe*
*ctrum*, 10(4), e01347-22. <https://doi.org/10.1128/spectrum.01347-22>.

Yang, L. N., Ren, M., Zhan, J. (2023). Modeling plant diseases under climate change: evolutionar
y perspectives. *Trends in Plant Science*, 28(5), 519-526. [https://doi.org/10.1016/j.tplants.202](https://doi.org/10.1016/j.tplants.2022.12.011)
[2.12.011](https://doi.org/10.1016/j.tplants.2022.12.011).

Yao, H., Sun, X., He, C., Maitra, P., Li, X. C., Guo, L. D. (2019). Phyllosphere epiphytic and endo
phytic fungal community and network structures differ in a tropical mangrove ecosystem. *Mi*
*crobiome*, 7, 1-15. <https://doi.org/10.1186/s40168-019-0671-0>.

Yin, Y., Wang, Y. F., Cui, H. L., Zhou, R., Li, L., Duan, G. L., Zhu, Y. G. (2023). Distinctive struct
ure and assembly of phyllosphere microbial communities between wild and cultivated ric
e. *Microbiology Spectrum*, 11(1), e04371-22. <https://doi.org/10.1128/spectrum.04371-22>.

Yue, H., Yue, W., Jiao, S., Kim, H., Lee, Y. H., Wei, G., Song, W., Shu, D. (2023). Plant domestica
tion shapes rhizosphere microbiome assembly and metabolic functions. *Microbiome*, 11(1), 1
725 -19. <https://doi.org/10.1186/s40168-023-01513-1>.

- Zhang, J., Zhang, B., Liu, Y., Guo, Y., Shi, P., Wei, G. (2018). Distinct large-scale biogeographic p
atterns of fungal communities in bulk soil and soybean rhizosphere in China. *Science of the t*
*otal environment*, 644, 791-800. <https://doi.org/10.1016/j.scitotenv.2018.07.016>.
- Zhang, L., Zhang, M., Huang, S., Li, L., Gao, Q., Wang, Y., ... & Ai, C. (2022). A highly conserve
730 d core bacterial microbiota with nitrogen-fixation capacity inhabits the xylem sap in maize pl
ants. *Nature communications*, 13(1), 3361.
- Zheng, Y., Maitra, P., Gan, H. Y., Chen, L., Li, S., Tu, T., Chen, L., Mi, X., Gao, C., Zhang, D., Gu
o, L. D. (2021). Soil fungal diversity and community assembly: affected by island size or typ
e? *FEMS Microbiology Ecology*, 97(5), fiab062. <https://doi.org/10.1093/femsec/fiab062>.
- Zhu, C., Lin, Y., Wang, Z., Luo, W., Zhang, Y., Chu, C. (2023). Community assembly and network
structure of epiphytic and endophytic phyllosphere fungi in a subtropical mangrove ecosyste
737 m. *Frontiers in Microbiology*, 14, 1147285. <https://doi.org/10.3389/fmicb.2023.1147285>.
- Zhou, J., Ning, D. (2017). Stochastic community assembly: does it matter in microbial ecolog
y? *Microbiology and molecular biology reviews*, 81(4), 10-1128. <https://doi.org/10.1128/mmbr.00002-17>.
- Zhou, Y., Jia, X., Han, L., Tian, G., Kang, S., & Zhao, Y. (2021). Spatial characteristics of the dom
inant fungi and their driving factors in forest soils in the Qinling Mountains, China. *Catena*, 2
06, 105504. <https://doi.org/10.1016/j.catena.2021.105504>.

**Supplementary figures**

**Figure S1** Study sites on Hainan Island. Red and blue solid circles each represent *V.*
*mangachapoi* and *D. pectinatum* on the map.

**Figure S2** Sampling design on Hainan Island. For each plot, we selected three trees
for sampled root (yellow solid circles) and leaves (green solid circles) from all four
cardinal directions.

**Figure S3** Fungal community composition at class level of *V. mangachapoi* and *D.*
*pectinatum* in the different compartments.

**Figure S4** Fungal sources of different compartments of *D. pectinatum* and *V.*
*mangachapoi*.

**Figure S5** The relative importance of different ecological processes in dominant
fungal class.

**Supplementary tables**

**Table S1** Leaves and soil physicochemical properties of *D. pectinatum* and *V.*
*mangachapoi* in different geographical locations.

**Table S2** Leaves and soil physicochemical properties for different tree species.
Different lowercase letters in the same row indicated significant differences ($P <$
0.05).

**Table S3** Multivariate analysis of variance results on the effects of geographical
locations, plant compartments and plant identity on α -diversity (OTU richness) of
fungi in *D. pectinatum* and *V. mangachapoi*.

**Table S4** Permuted multivariate analysis of variance (PERMANOVA) tables for
differences in fungal community compositions (OTU level).

**Table S5** The classification distribution results of OTUs at different classification
levels.

**Figure 1:** Diversity of fungal community in *D. pectinatum* and *V. mangachapoi*. A:
 Accumulated OTUs (γ -diversity) for all samples in five compartments; B: Mean OTU
 richness (α -diversity) of fungal communities for five compartments. C: Comparison
 of OTU richness of five compartments of *D. pectinatum* and *V. mangachapoi*. D:
 Comparison of OTU richness of *D. pectinatum* across five compartments in three
 locations. E: Comparison of OTU richness of *V. mangachapoi* across four
 compartments in four locations. Compartment abbreviation: LE, Leaf endophytic; PP,
 Leaf epiphytic; RP, Rhizoplane; RS, Rhizosphere; RE, Root endosphere; D, *D.*
 *pectinatum*; V, *V. mangachapoi*; DL, Diaoluo; JF, Jianfeng; WZ, Wuzhi; BW,
 Bawang; WN, Wanning; Significance level: * $p < 0.05$; ** $p < 0.01$; *** $p < 0.001$.

Figure 2 The fungal community compositions and the composition of the fungal community functional groups inferred by FUNGuild for *D. pectinatum* and *V. mangachapoi*. A: Phylum level; B: Class level; C: Relative abundance fungal functions; D: Relative abundance of fungal functions of dominant fungal class. Compartment abbreviation: Ag: Aboveground compartment; Bg: Belowground compartment. Significance level: * $p < 0.05$; ** $p < 0.01$; *** $p < 0.001$.

Figure 3 Principal coordinates analysis (PCoA) of taxonomic similarity based on

Bray–Curtis distances for fungal community compositions at the OTU level. A: Five
compartments. B: *D. pectinatum* and *V. mangachapoi* in five geographical locations.
Compartment abbreviation: LE, Leaf endophytic; PP, Leaf epiphytic; RP, Rhizoplane;
RS, Rhizosphere; RE, Root endosphere; D, *D. pectinatum*; V, *V. mangachapoi*; DL,
Diaoluo; JF, Jianfeng; WZ, Wuzhi; BW, Bawang; WN, Wanning; Significance level:
* $p < 0.05$; ** $p < 0.01$; *** $p < 0.001$.

 **Figure 4** Pairwise comparisons of the environmental factors are shown. Composition
 (OTU level) and diversity (observed OTU richness) of community of different
 compartments of *D. pectinatum* and *V. mangachapoi* was related to each
 environmental factor by partial Mantel tests. Edge width corresponds to the Mantel's r
 statistic for the corresponding distance correlation, and edges color denotes the
 statistical significance based on 999 permutations. Compartment abbreviation: LE,
 Leaf endophytic; PP, Leaf epiphytic; RP, Rhizoplane; RS, Rhizosphere; RE, Root
 endosphere. Significance level: * $p < 0.05$; ** $p < 0.01$; *** $p < 0.001$.

Figure 5 The relationship among environmental factors, the diversity (Sobs index), and the relative abundance of *Eurotiomycetes* in leaf and *Agaricomycetes* in root in *D. pectinatum* and *V. mangachapoi*. A: The relationship between environmental factors and Sobs index of leaf. The light green solid circles represent samples of Bawang, and dark green is Jianfeng. The light brown solid circles represent samples of Diaoluo, and dark brown is Wuzhi. The dark gray solid circles represent samples of Wanning. B: The relationship between environmental factors and the relative abundance of *Eurotiomycetes* in leaf. C: The relationship between environmental factors and the relative abundance of *Agaricomycetes* in root. The dark green solid circles represent samples of *D. pectinatum*, and dark brown is *V. mangachapoi*. Significance level: * $p < 0.05$; ** $p < 0.01$; *** $p < 0.001$.

**Fig 6** Assembly processes of the fungal community in five compartments. A and B:
 the boxplots of β -nearest taxon index (β NTI) and Raup-Crick based Bray-Curtis for
 all pairs of communities in five plant compartments. C: the relative importance of
 different ecological processes in five plant compartments. Compartment abbreviation:
 LE, Leaf endophytic; PP, Leaf epiphytic; RP, Rhizoplane; RS, Rhizosphere; RE, Root
 endosphere.

Manuscript ID: Spectrum03092-24

Title: The Distinct Diversity Patterns of Fungal Communities in Aboveground and Belowground Compartments of Tropical Tree Species

Journal title: Microbiology Spectrum

Dear Reviewer,

I hope this email finds you well. I am writing to express my gratitude for your assistance in reviewing my manuscript titled “The Distinct Diversity Patterns of Fungal Communities in Aboveground and Belowground Compartments of Tropical Tree Species” (03092-24). I sincerely appreciate the time and effort you and the expert reviewers have dedicated to providing invaluable feedback and suggestions.

After carefully considering the expert advice and suggestions, I have made significant revisions to the manuscript. I believe these changes have significantly strengthened the quality and clarity of the content. I have now re-uploaded the Revised manuscript, which incorporates the recommended changes, and I would be grateful if you could kindly reassess it.

I want to take this opportunity to express my sincere appreciation for your guidance throughout this process. The valuable insights and recommendations provided by you and the expert reviewers have been instrumental in improving the overall quality of the manuscript. Your expertise and thorough evaluation have immensely contributed to the enhancement of my work.

Once again, thank you for your dedicated efforts and commitment to supporting my research. I am truly grateful for your assistance. Should you have any further suggestions or require additional information, please do not hesitate to let me know.

Looking forward to hearing from you.

Best regards,

Kepeng Ji

Comments and Suggestions for Authors

The Revised parts will be marked in red font in the manuscript.

Reviewer 1

The manuscript utilizes high-throughput sequencing technology to analyze the diversity and function of microbial communities in the rhizosphere and phyllosphere of two dominant tropical rainforest tree species, *Dacrydium pectinatum* and *Vatica mangachapoi*. This research is of great significance for the conservation of tropical rainforests. The paper is well-written with detailed methodologies.

1. It is recommended to change the title of the paper to "Distribution Patterns of Fungal Community Diversity in the Dominant Tree Species *Dacrydium pectinatum* and *Vatica mangachapoi* in Tropical Rainforests." Since the analysis focuses on only these two tree species, reflecting this in the title would be appropriate.

Response: Revised the title of the paper. Thanks again. L1-L3

2. (Lines 25-27): Propose to be modified as: To address this research gap, high-throughput sequencing technology was performed to investigate the diversity of fungal communities in leaves and roots compartments of *D. pectinatum* and *V. mangachapoi* from Hainan Island of China.

Response: Revised. Thanks again. L26-L29

3. (Lines 25-26) In the abstract, "Aboveground compartments" should be Revised to "leaves," and "belowground compartments" should be changed to "roots" to enhance reader comprehension.

Response: Revised. Thanks again. L29-L37

4. In addition, for a research article no data were exhibited in the whole abstract, which was likely improper. Advise to supplement some key data results.

Response: Added. Thanks again. L29-L32; L34-L37

5. The references in the introduction section of the article need to be updated.

Response: The references in the introduction section have been updated. L492; L512; L514; L554

6. In the introduction, it is suggested to add one or two sentences briefly describing the basic characteristics of Hainan's tropical rainforest to highlight its importance in global biodiversity conservation.

Response: Added. Thanks again. L96-100

7. (Lines 118): Propose to be modified as: Description of study area and site selection.

Response: Revised. Thanks again. L128

8. (Lines 131): Change the 2.2 subheading to "Sample collection and analysis".

Response: Revised. Thanks again. L141

9. (Lines 347-348): The abstract of the article mentions that soil pH mainly affects underground microorganisms, which needs to be discussed in this section to support your point.

Response: Added. Thanks again. L372-376

10. (Lines 437-440): Propose to be modified as: In summary, fungal communities in leaves and roots compartments exhibited significant variation in their diversity between *Dacrydium pectinatum* and *Vatica mangachapoi*, and the factors driving these communities were distinct.

Response: Revised. Thanks again. L441-444

11. The vertical axis labels in Figures 5B and 5C should be "Abundance of Eurotiomycetes" or "Agaricomycetes."

Response: Revised in Figures 5B and 5C. Thanks again.

12. Please, be sure that all the references cited in the manuscript are also included in the reference list and vice versa with matching spellings and dates.

Response: Revised. Thanks again.

Reviewer 2

The manuscript by Ji and co-authors describes a detailed metabarcoding analysis of the fungal communities of two tropical trees, *Dacrydium pectinatum* and *Vatica mangachapoi*, in different above- and below-ground compartments. Samples were obtained from seven different locations (four for *V. mangachapoi*, three for *D. pectinatum*) and in total 105 samples were analysed. The study provides interesting insight in the fungal communities in different plant and soil compartments and compares also across sites. The authors also performed analyses to predict the most important factors affecting community structure and assembly. Overall, it is an interesting and detailed analysis. A few general comments and suggestions for improvements are given below. Detailed corrections are directly suggested in the attached Word document (in track-change mode).

1. Lines 142f: Please specify the buffered solution.

Response: Added. Thanks again. "To obtain leaf surface microbiota samples, leaves were placed in a sterile buffered solution (0.1 M Potassium Phosphate, 0.1% Glycerol, 0.15% Tween 80, pH 7.0)." L151-152

2. Line 148ff: Please provide more information about the collection of the soil samples.

Response: Added. L158-166

3. Section 2.3: All words at the end of the line are split in a strange manner.

Response: Revised. Thanks again. L168-180

4. Line 192: Maybe provide some information and reference for the Sobs diversity index.

Response: Added. L199-200

5. Section 3.2: Would it not be possible to provide more information on a lower taxonomic level? Ascomycota and Basidiomycota dominate virtually all compartments and niches anywhere. This is not informative. Maybe an analysis of the overlapping and not overlapping taxa would provide valuable information? Often people show Venn diagrams for this purpose. The authors may even analyze their data to see if a kind of core mycobiome could be defined.

Response: We analyzed the shared taxa at the genus and species levels using Venn diagrams, and some results are presented in the Supporting Information Figure S4. However, due to the incomplete annotation and alignment of fungi in the sequencing database, the majority of the results remain unidentified. L267-272

6. Line 281: What is AN?

Response: The information of Ammonium Nitrogen (AN) has already been presented in the Supporting Information. L22

7. Line 313: Is the diversity of bacteria also higher aboveground than in soil? This reviewer would have expected a higher diversity in soil.

Response: In the study that we have published, the bacterial α -diversity in the roots was higher than in the leaves. However, in this fungal study, the fungal α -diversity in the leaves is higher than in the roots.

8. Line 324: Does Figure 1C not suggest that plant identity does not affect the fungal community?

Response: Indeed, Figure 1C suggest that plant identity does not affect the fungal α -diversity. But fungal community composition was shaped by compartment, geographic location, and plant identity (Figure 3A).

9. Line 439 (first sentence of conclusion) is not complete.

Response: Revised. "We analyzed the diversity, composition and community assembly patterns of fungal communities in endangered trees." L440-L441

10. Figures: It would be preferable to change the colors for the different sites at the two locations. The shades of green and brown are very difficult to distinguish; in particular in Figures 2 and 3-5. Maybe it would also be possible to add labels for the above- and belowground samples in each graph to make this distinction easier to recognize (for example a labelled bar).

Response: I made the changes again. In Figure 2, I used the labels "Ag: Aboveground compartment" and "Bg: Belowground compartment" to distinguish between the two. In Figure 3-5, I also added labels for the above- and belowground samples in each graph to make this distinction easier to recognize.

11. Figure 2: It is not clear to this reviewer how the panels A and B are connected. There are a lot more Ascomycota in B than in A (based on percentage).

Response: Figure 2A represents Top 10 percentage of Phylum level and Figure 2B represents Top 10 percentage of Class level.

In simple terms:

The percentage of Ascomycota is calculated relative to the abundance of all phyla in the entire group.

The percentage of each class under Ascomycota is calculated relative to the abundance of all classes in the entire group.

Therefore, the percentage of Basidiomycota in Figure 2A is may lower than the sum of the percentages of all its classes in Figure 2B.

Re: Spectrum03092-24R1 (Distribution Patterns of Fungal Community Diversity in the Dominant Tree Species *Dacrydium pectinatum* and *Vatica mangachapoi* in Tropical Rainforests)

Dear Prof. Guoyu Lan:

Apologies for the delayed response to the submission of your revised manuscript, which has now been accepted and forwarded to the ASM production staff for publication. I was certain that I had accepted it already a while ago, but I must have made a mistake.

Your paper will first be checked to make sure all elements meet the technical requirements. ASM staff will contact you if anything needs to be revised before copyediting and production can begin. Otherwise, you will be notified when your proofs are ready to be viewed.

Sincerely,
Florian Freimoser
Editor
Microbiology Spectrum